# Elucidation of *Spartina* dimethylsulfoniopropionate synthesis genes enables engineering of stress tolerant plants

Rocky D. Payet [1], Lorelei J. Bilham [1], Shah Md Tamim Kabir [1], Serena Monaco [2], Ash R. Norcott [1], Mellieha G. E. Allen [1], Xiao-Yu Zhu [1], Anthony J. Davy [1], Charles A. Brearley [1], Jonathan D. Todd [1,3] ✉ & J. Benjamin Miller [1] ✉

The organosulfur compound dimethylsulfoniopropionate (DMSP) has key roles in stress protection, global carbon and sulfur cycling, chemotaxis, and is a major source of climate-active gases. Saltmarshes are global hotspots for DMSP cycling due to *Spartina* cordgrasses that produce exceptionally high concentrations of DMSP. Here, in *Spartina anglica*, we identify the plant genes that underpin high-level DMSP synthesis: methionine *S*-methyltransferase (*MMT*), *S*-methylmethionine decarboxylase (*SDC*) and DMSP-amine oxidase (*DOX*). Homologs of these enzymes are common in plants, but differences in expression and catalytic efficiency explain why *S. anglica* accumulates such high DMSP concentrations and other plants only accumulate low concentrations. Furthermore, DMSP accumulation in *S. anglica* is consistent with DMSP having a role in oxidative and osmotic stress protection. Importantly, administration of DMSP by root uptake or over-expression of *Spartina* DMSP synthesis genes confers plant tolerance to salinity and drought offering a route for future bioengineering for sustainable crop production.

Dimethylsulfoniopropionate (DMSP) is one of Earth's most abundant and ecologically important organosulfur molecules, with roles as an osmolyte[1], cryoprotectant[2], antioxidant[3], baroprotectant[4], grazing deterrent[5], in chemotaxis[6], global carbon and sulfur cycling, and, potentially, in climate regulation, since it is a major precursor for the climate-active gases dimethylsulfide (DMS) and methanethiol[7]. Marine algae and bacteria are major global DMSP producers[8], but invasive perennial *Spartina* grasses[9] (e.g., *S. anglica* and *S. alterniflora*) also display high intracellular DMSP concentrations and are responsible for considerable DMSP production and cycling in saltmarshes. Per unit area, the DMSP concentrations produced by these plants are far higher than the oceans and are proposed to contribute ~10% of global atmospheric DMS emissions[10]. Only seagrasses such as *Posidonia oceanica* and *Zostera marina*[11], *Saccharum officinarum* (sugarcane)[12], and sea daisy *Melanthera biflora* (formerly *Wollastonia biflora*)[13] produce similarly high DMSP concentrations, though some other plants can produce DMSP at concentrations 1000 times lower[14,15]. Not all *Spartina* spp. produce DMSP at high concentrations (e.g., *S. patens* and *S. versicolor*) implying that phylogenetic relatedness is likely a poor indicator of DMSP production in plants, but a coordinated study of the plant kingdom is needed.

Here, we identify the genes that underpin high-level DMSP synthesis in plants. We show that *S. anglica* DMSP synthesis enzymes are efficient and highly expressed compared with homologs from plants that accumulate low DMSP concentrations. We demonstrate that DMSP has roles in oxidative and osmotic stress, that DMSP levels

[1]School of Biological Sciences, University of East Anglia, Norwich Research Park, Norwich NR4 7TJ, UK. [2]School of Chemistry, Pharmacy, and Pharmacology, University of East Anglia, Norwich Research Park, Norwich NR4 7TJ, UK. [3]MOE Key Laboratory of Evolution and Marine Biodiversity, Frontiers Science Center for Deep Ocean Multispheres and Earth System & College of Marine Life Sciences, Ocean University of China, Qingdao 266003, China. ✉e-mail: jonathan.todd@uea.ac.uk; b.miller@uea.ac.uk

are manipulable in low-accumulating species and that doing so can increase tolerance to salinity and drought. This work provides molecular genetic understanding of DMSP production in plants. The findings open new research avenues to address the contribution of plants to global DMSP cycling and identifies a role for DMSP in plant biotechnology.

## Results

### Identification and ratification of DMSP synthesis genes from *Spartina anglica*

We first set out to identify candidate DMSP synthesis genes in *Spartina anglica* (Fig. 1a), which is xenonative to the UK, an invasive coloniser of coastal wetlands worldwide[9], and hosts key DMSP catabolic bacteria[16]. *Spartina* produces DMSP from the amino acid L-methionine via a methionine methylation pathway[17] (Fig. 1b) that comprises four sequential enzymes: Methionine *S*-methyltransferase (termed MMT), *S*-methylmethionine (SMM) decarboxylase (termed SDC), DMSP-amine oxidase (termed DOX) and DMSP-aldehyde dehydrogenase (termed ALDH). Analysis of mRNA sequencing from triplicate *S. anglica* leaves, which contained 6 µmol g$^{-1}$ FW DMSP, identified two transcripts from different loci that encoded MMT proteins, SaMMT1 and SaMMT2. These proteins shared >84% amino acid identity to the MMT enzyme from *Zea mays*. MMT is ubiquitous in plants[18], catalyses the production of SMM as part of the SMM cycle[19] and is the first step of plant DMSP synthesis (Fig. 1b). SaMMT2, which contained a 10 amino acid insertion at the conserved *S*-adenosylmethionine-binding domain (Supplementary Fig. 1), showed no MMT activity and, thus, is not involved in SMM cycling or DMSP production. In contrast, SaMMT1, which contained no insertion in this domain, readily demonstrated in vitro MMT activity (Supplementary Fig. 1). Using the *S. anglica* transcriptome, we also identified 11 transcripts (Supplementary Data 1) encoding candidate aldehyde dehydrogenases highly homologous (75% protein identity) to characterised enzymes, notably betaine aldehyde dehydrogenase (BADH1 and BADH2) that were previously shown to have significant ALDH activity[20]. However, since aldehyde dehydrogenases are promiscuous in their substrate range, readily convert DMSP-aldehyde into DMSP in species that do not accumulate DMSP to high concentrations[20], and (like MMT) are ubiquitous in plants, neither MMT nor ALDH are indicative of high-level production of DMSP in plants. This left the *Spartina* SDC and DOX enzymes as specific to plant DMSP synthesis[17] and as the best candidates for determining high-level DMSP production, especially as neither enzyme had been identified in a plant.

The *S. anglica* transcriptome was screened for genes encoding amino acid decarboxylases and copper amine oxidases, since SDC and DOX were previously predicted to belong to these enzyme families, respectively[17]. This identified three candidate SDC enzymes predicted to encode an arginine decarboxylase (SaADC), ornithine decarboxylase (SaODC) and diaminopimelate decarboxylase (SaDAPDC) (Supplementary Data 2). SaDAPDC shared the highest identity to bacterial SMMDC and Burl proteins (with 27% and 24% amino acid identity, respectively) that catalyse this reaction in DMSP producing bacteria[21,22]. When expressed in and purified from *E. coli*, SaADC and SaDAPDC both showed in vitro activity with their predicted native substrates (arginine and diaminopimelate, respectively; Supplementary Fig. 2a). However, when provided with SMM, only SaODC demonstrated SDC activity (Fig. 1c; Supplementary Fig. 2b, c), showing a $K_m$ of 2.13 ± 0.20 mM and a $K_{cat}$ of 1.67 ± 0.08 nmol µg$^{-1}$ min$^{-1}$ protein (Supplementary Fig. 2d). Contrastingly, SaODC demonstrated virtually no activity with ornithine (Supplementary Fig. 2e, f). Additionally, SMM was roughly 14-times more abundant than ornithine in *S. anglica* leaf extracts. Therefore, we subsequently refer to the SaODC enzyme as SaSDC.

Two candidate DOX enzymes (copper amine oxidases) were identified in *S. anglica* that shared 30% amino acid identity and were termed SaCAO1 and SaCAO2 (Supplementary Data 2). Recombinant SaCAO1 and SaCAO2 proteins were assayed for DOX activity, using pig kidney diamine oxidase as a positive control[17]. Both SaCAO1 and SaCAO2 showed DOX activity, but the latter was ~45-fold more active (Fig. 1d), with an approximate $K_{cat}$ of 298.57 pmol µg$^{-1}$ min$^{-1}$ (compared with 6.63 pmol µg$^{-1}$ min$^{-1}$ for SaCAO1). Furthermore, these enzymes were both independently identified by mass spectrometry analysis of the most active protein fractions from chromatographic experiments that enriched for *S. anglica* DOX activity (Supplementary Fig. 3; Supplementary Data 3), consistent with them (particularly SaCAO2) being the enzymes responsible for DOX activity in this plant. As well as DMSP amine, SaCAO2 could use putrescine as a substrate and its high DOX activity was decreased by 97% in the presence of 20-fold excess putrescine, consistent with what is seen in *S. anglica* protein extracts (Fig. 1d), and previous work on DOX activity in *Spartina* extracts[17]. SaCAO1 showed no activity towards putrescine. We therefore refer to SaCAO2 as SaDOX for the remainder of this study.

We next assessed the expression of *SaMMT1*, *SaCAO1*, *SaDOX* and *SaSDC* across different *S. anglica* tissues by RT-qPCR. Transcript levels of these genes, but most prominently, *SaSDC*, were significantly higher in leaves than any other tissue type (Fig. 1e). Furthermore, SDC and DOX enzyme activities were also both highest in leaves (Fig. 1f, g), where plant DMSP synthesis was previously reported to occur[23]. Indeed, *SaSDC* transcripts and corresponding SDC enzyme activity were virtually undetectable or undetected, respectively, in any tissue besides leaf. Despite this, DMSP was still detected at high concentrations in other tissues, at 16–58% of the concentration in leaves (Fig. 1h). Since SDC transcripts and activity were exclusively found in leaves (Fig. 1e, f), our findings are consistent with DMSP being synthesised in the leaves and then subsequently mobilised throughout the plant to other tissues. It is also possible that some DMSP-amine synthesised in leaf tissue was exported to other tissues where it could be converted to DMSP, since DOX activity was only 21–42% decreased in non-leaf extracts (Fig. 1g). Whether this non-leaf DOX activity is due to SaCAO1, SaDOX and/or other unknown enzymes requires future investigation.

### DMSP accumulation in *Spartina anglica* is associated with abiotic stress

*Spartina*-populated saltmarshes are significant sources of climate-active gas emissions via DMSP catabolism[10], but little is currently known about how synthesis and accumulation of DMSP varies across natural *Spartina* populations, or what regulates DMSP levels in them. To address this, we measured DMSP concentrations in leaf samples from clumps of *S. anglica* growing across a transect in the saltmarsh at Stiffkey, Norfolk, UK (Fig. 2a; Supplementary Fig. 4a). Our sampling strategy targeted a natural population across a single saltmarsh, which represents relatively limited genetic diversity (since *S. anglica* primarily undergoes vegetative propagation and the species formed from a hybridisation event some 200 years ago, resulting in a genetic bottleneck[24]), thus allowing us to focus on uncovering environmental (rather than genetic) factors that control DMSP accumulation. Unexpectedly, DMSP accumulation was highly variable, with differences as great as 14-fold (Fig. 2b). This is particularly striking given the magnitude of DMSP levels (> 0.4–6 µmol g$^{-1}$ FW). Moreover, there was no correlation in DMSP concentrations between clonal clumps growing within the same sampling site.

RNA-seq and differential expression analysis of the three lowest and three highest DMSP accumulating clumps (Supplementary Data 4; Supplementary Fig. 4b) revealed the highest DMSP accumulating plants to have strongly elevated expression of an ethylene-responsive transcription factor 1 (*EREBP1*) homolog (Fig. 2c), a transcription factor whose expression has been linked to drought, submergence and pathogen stresses in other grasses[25,26]. This upregulation suggested that these highest DMSP accumulating plants were experiencing elevated levels of one of these stresses, and concurrently had elevated

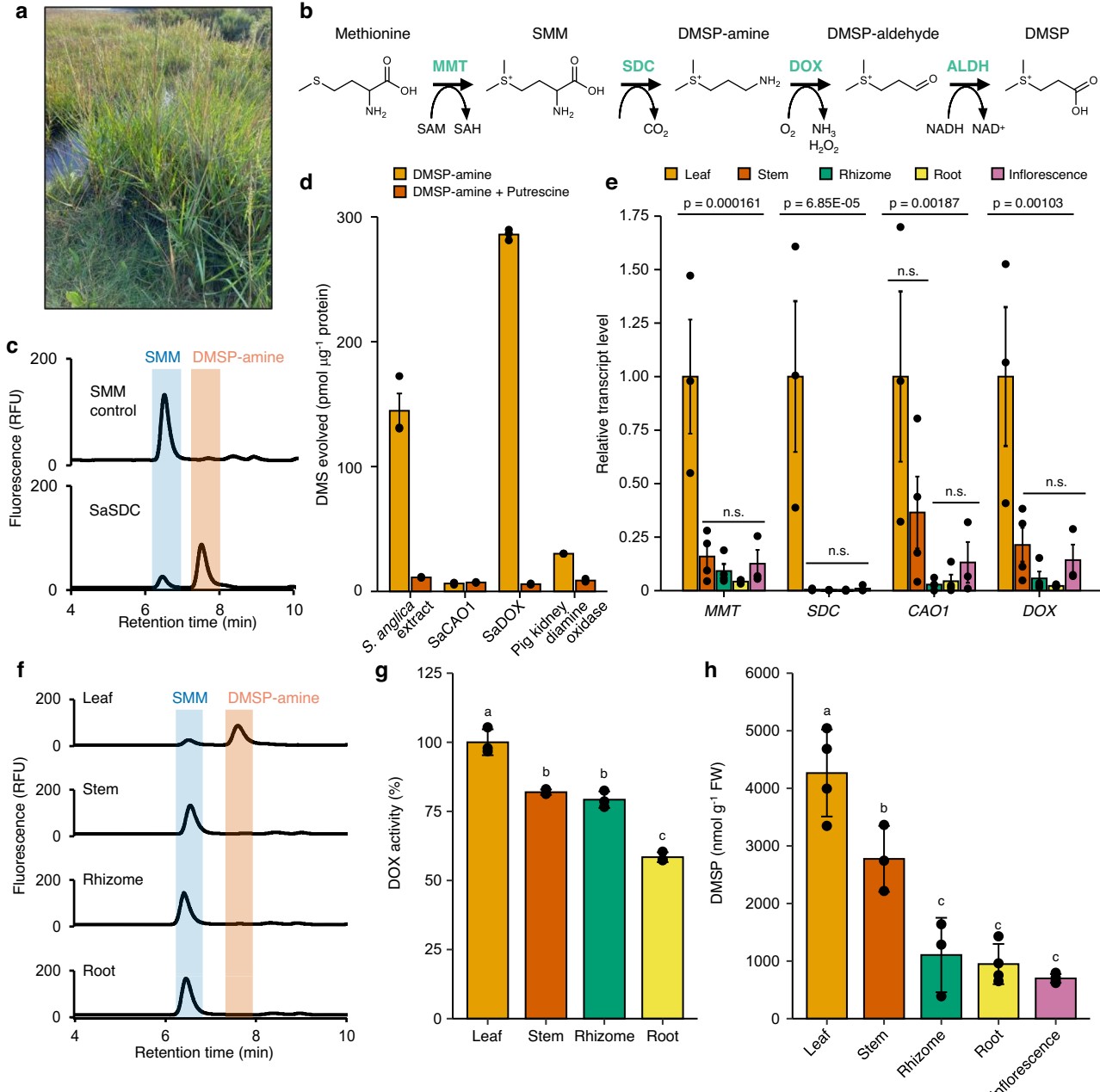

**Fig. 1 | *Spartina anglica* produces DMSP via SDC and DOX. a** *Spartina anglica* at Stiffkey saltmarsh, May 2021. **b** DMSP synthesis pathway in *S. anglica* converts methionine to DMSP via four enzyme activities (green). **c** HPLC trace of SDC assay for recombinant *S. anglica* SDC (SaSDC, *S*-methylmethionine decarboxylase) incubated with SMM, leading to production of DMSP-amine, alongside an SMM-only control. **d** GC result of DOX assay performed on enzyme extracts from *S. anglica* leaf tissue, recombinant *S. anglica* copper amine oxidase 1 (SaCAO1) and recombinant *S. anglica* DOX (SaDOX, DMSP-amine oxidase). Enzymes were incubated with DMSP-amine or DMSP-amine and competing substrate putrescine in excess, leading to production of DMSP-aldehyde that spontaneously degrades to produce DMS. Pig kidney diamine oxidase used as a positive control ($n = 3$ per protein and substrate). **e** RT-qPCR of *S. anglica* DMSP synthesis genes in denoted tissues ($n = 4$ for leaf, rhizome, root and stem, $n = 3$ for inflorescence). *p*-values indicate statistical significance after one-way ANOVA followed by Tukey test; n.s.

denotes no statistically significant difference. **f** HPLC traces of SDC assays performed on enzyme extracts prepared from denoted *S. anglica* tissues. Retention times of SMM (6.5 min) and DMSP-amine (7.6 min) are indicated in blue and red, respectively. **g** DOX assays performed on enzyme extracts prepared from denoted *S. anglica* tissues, normalised to 1 μg of total extracted protein and with activities shown relative to leaf ($n = 3$, $p = 0.00000373$). Letters indicate statistical significance after one-way ANOVA followed by Tukey test. **h** DMSP accumulation in denoted *S. anglica* tissues ($n = 4$ for inflorescence, leaf and root, $n = 3$ for rhizome and stem, $p = 0.041686$ for leaf vs stem, $p = 0.005$ for leaf vs rhizome, $p = 0.00002820$ for leaf vs root, $p = 0.0000584$ for leaf vs inflorescence). Letters indicate statistical significance after one-way ANOVA followed by Tukey test. Data represent mean ± one standard error. Source data are provided as a Source Data file.

concentrations of DMSP; DMSP synthesis may also be regulated by EREBP1. Expression of *MMT*, *SDC* and *DOX* did not correlate with standing stocks of DMSP (Supplementary Fig. 4c), suggesting that these genes may be post-translationally regulated, that DMSP is further metabolised[27] or that DMSP synthesis is influenced by substrate partitioning/availability[28]. Gene ontology enrichment analysis of differentially expressed genes (Fig. 2d) demonstrated a significant enrichment in high accumulators for biological process terms relating

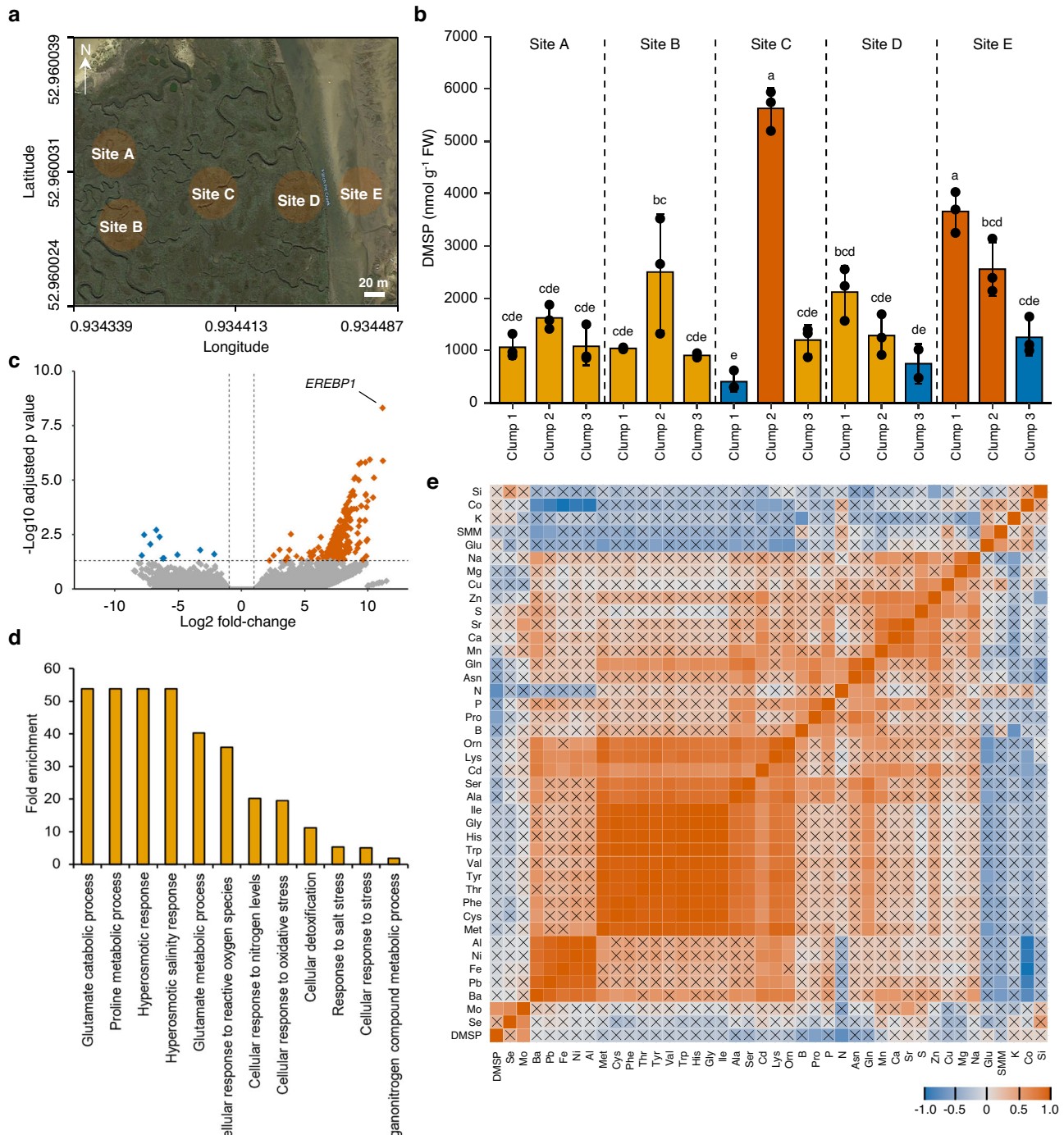

**Fig. 2 | DMSP accumulation in *Spartina anglica* is associated with stress responses. a** Satellite image of *Spartina anglica* sampling sites at Stiffkey salt-marsh, taken from Google Maps. **b** Variability in DMSP accumulation in individual clumps of *S. anglica* harvested from the sites denoted in panel a. Samples identified as 'highest accumulators' (red) and 'lowest accumulators' (blue) were subjected to RNA-sequencing ($n = 3$ per clump per site, $p = 8.3e\text{-}14$ in one-way ANOVA). Letters indicate statistical significance after one-way ANOVA followed by Tukey test. **c** Volcano plot demonstrating contrasting gene expression profiles between highest and lowest DMSP accumulating clumps of *S. anglica*. Each dot represents a gene, with red dots indicating genes that are more highly expressed in highest accumulators, blue dots indicating genes that are lower, and grey dots denoting genes that are not significantly differentially expressed. Dotted lines indicate

significance thresholds from DESeq2 analysis. Labelled dot refers to Ethylene Responsive Element Binding Protein 1 (*EREBP1*), a key stress responsive transcription factor in grasses. **d** Gene ontology enrichment analysis of differentially expressed genes from (**c**), demonstrating an enrichment of stress-responsive genes in highest DMSP accumulators. **e** Heatmap of pairwise Pearson correlations between measured variables (abundance of indicated elements and amino acids in *S. anglica* leaves), demonstrating negative linear relationship between DMSP and levels of nitrogen and proline, and positive linear relationship between DMSP and molybdenum (Mo). Crosses indicate that a relationship is not significant. Data represent mean ± one standard error. Source data are provided as a Source Data file.

to oxidative stress and reactive oxygen species, salt and osmotic stress, and cellular detoxification, in agreement with the hypothesis that DMSP synthesis is stress regulated[29]. There was also a substantial enrichment of biological process terms relating to the metabolism of proline (Fig. 2d), a known osmolyte in plants[30], suggesting a coupling with DMSP synthesis.

To confirm this connection with proline metabolism, we measured proline, glutamate, and ornithine concentrations in *S. anglica* leaf tissues, in conjunction with a selection of other amino acids to contextualise DMSP accumulation within primary metabolism. We also assayed a variety of nutrients (N, S, K, P, Mg, Se, Mo, Fe, Zn) and ions and heavy metals known to contaminate saltmarshes (Na, Cd, Ni, Cu, Pb, Sr, Mn, Ba), both in leaf tissue (Fig. 2e) and soil (Supplementary Fig. 4c), to determine if variation in these parameters influenced DMSP accumulation in the environment. Consistent with literature[31], a negative correlation was observed between nitrogen levels and DMSP. We found that proline also negatively correlated with DMSP (correlation of proline with nitrogen was just short of the *p*-value cut-off; $p = 0.08$) (Fig. 2e). Leaf DMSP content also negatively correlated with soil nitrogen (Supplementary Fig. 4d). Interestingly, DMSP positively correlated with the levels of molybdenum in leaves (Fig. 2e). This effect appears to be unrelated to the amount of molybdenum (Mo) in soil, as this relationship is lost when correlated with total soil Mo (Supplementary Fig. 4d). This is likely related to nitrogen deficiency, as Mo is used as a cofactor for nitrate reductase[32], and nitrogen uptake is a demand driven process. It is also possible that *S. anglica* sulfate transporters contribute to Mo accumulation, as in other plants[33]. Consistent with this, *S. anglica SULTR1;5*, *SULTR2;2* and *SULTR 2;6* showed expression in the root (Supplementary Fig. 5a, b), whilst the highest DMSP-accumulating plants had higher expression levels of *SULTR3;4*, *SULTR3;6* and *SULTR3;7* (Supplementary Fig. 5c). There is a strong positive relationship between methionine and most other core amino acids (Fig. 2e), as can be expected given their connection via primary metabolism. DMSP does not share a relationship with these amino acids, indicating its decoupling from primary metabolism, which is also seen for SMM. We also observed no relationship between methionine, SMM and DMSP levels (Fig. 2e), despite the former two being synthesis intermediates. This suggests that the regulation of flux occurs at a point after MMT, and that these amino acid pools are resilient to depletion by large fluxes of DMSP synthesis in *S. anglica*.

To test how DMSP accumulation is regulated by salt or sulfur status, we performed experiments with glasshouse-grown *S. anglica* plants watered with freshwater, salt solution (NaCl), or sea salts (containing NaCl and high concentrations of sulfate). Interestingly, DMSP accumulation was not significantly increased in plants watered with NaCl relative to plants treated with freshwater (consistent with[31,34]), but plants watered with sea salts did contain higher DMSP concentrations (Supplementary Fig. 5d). This finding suggests that DMSP accumulation is not regulated by NaCl treatment, but rather is regulated by sulfate availability. Consistent with this, in the natural *S. anglica* populations we found no correlation between DMSP accumulation and Na concentrations in the leaf (Fig. 2e) or soil (Supplementary Fig. 4d). Furthermore, measurements of sulfate and *O*-acetylserine (a key sulfur status indicator[35]) in leaf material from the three lowest and three highest DMSP accumulating clumps revealed no significant differences in the concentrations of these compounds (Supplementary Fig. 4e), suggesting that DMSP accumulation places no significant drain on the sulfur reserves of *S. anglica*. Overall, these findings support a role for DMSP synthesis being an adaptation of *S. anglica* to growing in a sulfur-rich saltmarsh environment.

Taken together, these data demonstrate that DMSP accumulates dynamically across populations of *S. anglica* in the environment. The level of accumulation depends on nitrogen availability, and the contrasted transcriptional profile of highest and lowest DMSP accumulators suggests a link with abiotic stress, supporting the role of DMSP as

an anti-stress molecule. DMSP accumulation is increased in response to elevated concentrations of sulfate but not salt. Additionally, the lack of correlation between transcript abundance of synthesis genes and standing stocks of DMSP suggest more complex regulation than simple up or downregulation of gene expression.

## DMSP synthesis is widespread in plants and evolution of high-level production is convergent

To determine the breadth of DMSP production in plants, DMSP concentrations were measured in phylogenetically and environmentally diverse species spanning the plant kingdom. Consistent with literature[14,15], all 43 tested plant species produced DMSP, albeit mostly at levels that were two or four orders of magnitude lower than in *Saccharum officinarum* or *Spartina anglica*, respectively (Fig. 3a). No new high-level DMSP accumulating plants were identified and this trait should therefore be considered rare. However, it should be noted that species were identified which accumulated DMSP concentrations only 5-times lower than *Saccharum officinarum*; these were *Araucaria araucana* (monkey puzzle; 125 nmol g$^{-1}$ FW) and *Eurhynchium striatum* (common striated feather-moss; 113 nmol g$^{-1}$ FW). DMSP accumulation at lower levels was universal in plants, implying that terrestrial systems may be a significant source of DMSP. There was no clear phylogenetic link between species that demonstrated high-level DMSP accumulation, indicating that this characteristic likely evolved convergently multiple times. Furthermore, DMSP concentrations were previously shown to be elevated in *Arundo donax* and *Solanum lycopersicum* in response to drought[14,15], suggesting that DMSP accumulation in other lower DMSP producing plants may also be enhanced by conditions linked to climate change and potentially become more significant sources of DMSP and related climate-active gases under such future conditions.

To determine how plants accumulate different concentrations of DMSP, we assessed the prevalence and activity of *Spartina* DMSP synthesis enzyme homologs from different species. SaMMT1 in vitro activity was comparable to previously characterised plant MMTs from *Arabidopsis thaliana* and *Hordeum vulgare*, which accumulate low concentrations of DMSP (Supplementary Fig. 1a). Phylogenetic analysis also showed plant MMT diversity to follow taxonomy, with MMT from high accumulating species not being distinct from those from low producers (Supplementary Fig. 6a).

High-level SDC activity was previously shown to be rare in plants[17]. Proteins homologous to *SaSDC* (50–84% amino acid identity) were also identified in most higher plants (Fig. 3b), but notably not in *A. thaliana*, which lacks ornithine decarboxylases[36], nor from the available RNA-seq data from the high DMSP producer *P. oceanica*[37] or in the genome sequence of *Zostera marina*, implying the existence of (an)other DMSP synthesis route(s) in these species. The proteins homologous to SaSDC from the high-level DMSP producer *Saccharum officinarum* and low-level DMSP producers *Solanum lycopersicum*, *Setaria viridis* and *Spartina patens*, all converted ornithine into putrescine (Supplementary Fig. 6b), but significantly, lacked SDC activity (Fig. 3c). Despite these stark differences in enzyme activity, SaSDC phylogenetically clustered with the other ODC enzymes in a manner reflective of phylogeny (Fig. 3b). Sequence analysis did not highlight any substantial amino acid insertions or deletions between SaSDC and those ODC enzymes lacking SDC activity to explain the gained activity of SaSDC on SMM substrate, implying this difference is likely due to a small number of specific amino acid substitutions. Together, these data imply that biochemical changes in SaODC that alter its function to that of an SDC, underlie evolution of high-level DMSP accumulation in *S. anglica*. We also conclude that *Saccharum officinarum* and seagrasses *P. oceanica* and *Z. marina*, must use distinct enzymes/pathways to synthesise and accumulate DMSP at high concentrations.

Proteins with 87–91% protein identity to SaDOX were ubiquitous in the plant kingdom (Fig. 3d) and, like MMT, showed much less

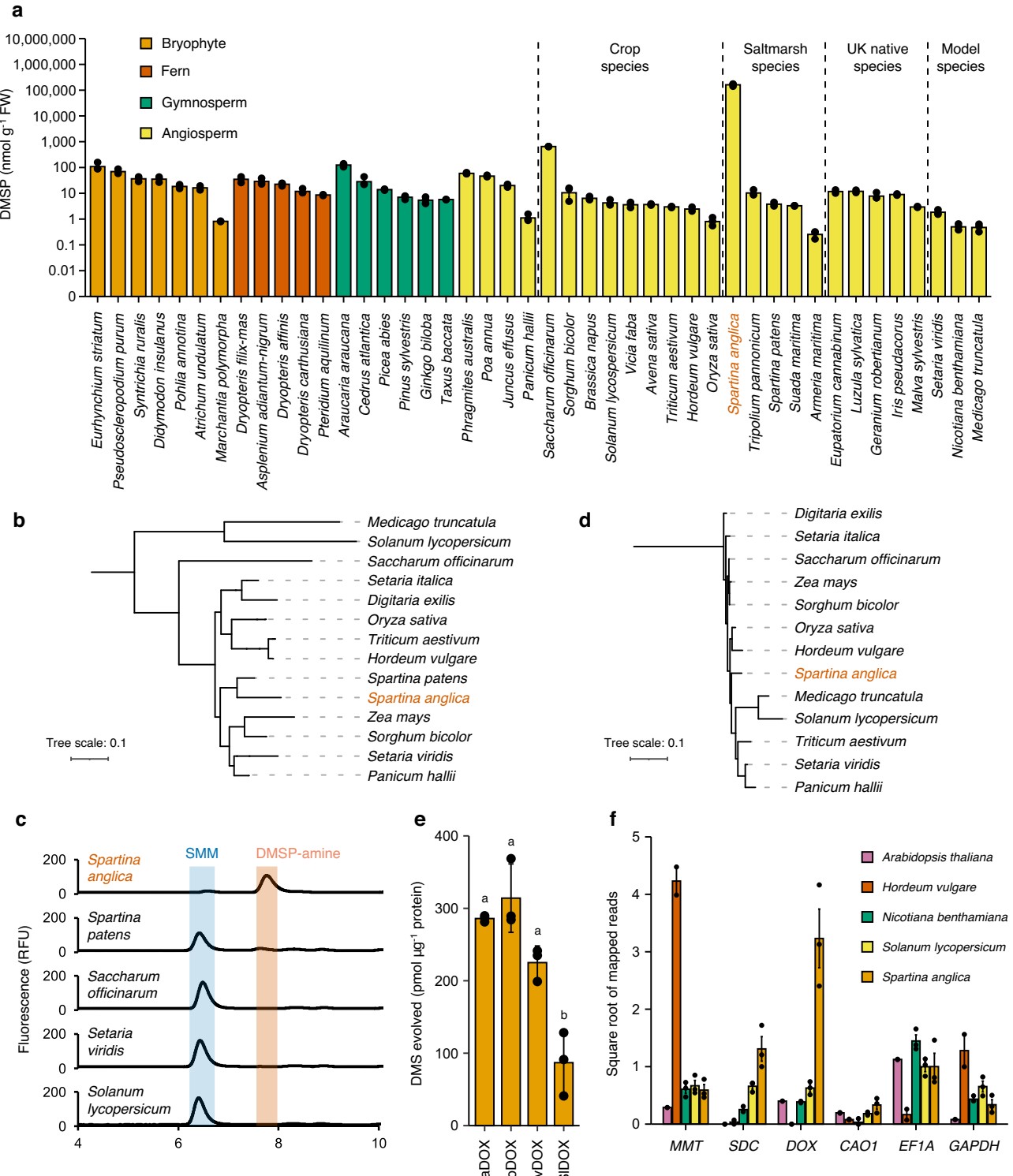

**Fig. 3 | *Spartina anglica* produces high concentrations of DMSP due to its unique SDC activity. a** DMSP measurements showing accumulation of the compound in a large diversity of different plant species (*n* = 3 per species). **b** Phylogenetic tree of ODC/SDCs in plant species related to *S. anglica*. **c** HPLC traces of SDC assays showing that the recombinant *S. anglica* SDC produces DMSP-amine from SMM, whilst enzyme homologs from other plant species do not have this activity. **d** Phylogenetic tree of DOX homologs in plant species related to *S. anglica*. **e** DOX assays performed with selected enzyme homologs from *S. anglica* (Sa), *Saccharum offinarum* (So), *Setaria viridis* (Sv) and *Solanum lycopersicum* (Sl) (*n* = 3 per recombinant enzyme, *p* = 0.7506270 for SaDOX vs SoDOX, *p* = 0.2102275 for SaDOX vs SvDOX, *p* = 0.0004608 for SaDOX vs SlDOX). Letters indicate

statistical significance after one-way ANOVA followed by Tukey test. **f** Number of reads mapping to homologs of denoted DMSP synthesis genes in each species, normalised to the number of reads mapping to *ACT8* homologs (square rooted for clarity), showing levels of *DOX* are much higher in *S. anglica*, despite equivalent enzymology. *EF1A* and *GAPDH* are included as reference gene controls, to demonstrate expression differences of other genes are not attributable to read-depth artefacts (*n* = 3 libraries for *Spartina anglica*, *Solanum lycopersicum* and *Nicotiana benthamiana*, *n* = 2 libraries for *Arabidopsis thaliana* and *Hordeum vulgare*). Data represent mean ± one standard error. Source data are provided as a Source Data file.

diversity than seen in the SDC/ODC family enzymes (Fig. 3b; Supplementary Fig. 6a). Indeed, SaDOX homologs from *S. officinarum* and *Setaria viridis* had DOX activity equivalent to the *S. anglica* DOX, and *Solanum lycopersicum* had lower activity (30%) (Fig. 3e), implying that enzymes with DOX activity were themselves not determinants of high-level DMSP accumulation. This seemingly contradicts previous work demonstrating enhanced DOX rates in *Spartina* species that accumulated high DMSP concentrations compared to plants now known to accumulate low DMSP concentrations[17]. We subsequently assayed protein extracts from *S. anglica*, *S. patens*, *Solanum lycopersicum*, *Nicotiana benthamiana*, *A. thaliana* and *H. vulgare* for DOX activity, species for which we were able to identify *DOX* homologs (with the exception of *Spartina patens*, for which no genetic resources exist and for which we were unable to clone a *DOX* based on sequences designed against *S. anglica DOX*). As reported[17], *S. anglica* showed much higher levels of DOX activity (12-times higher) than the low producing species (Supplementary Fig. 6c), and this activity was also inhibited by excess putrescine. To compare expression levels of the DMSP synthesis genes in these species, we analysed publicly available transcriptomic datasets (*A. thaliana*, *H. vulgare* and *N. benthamiana*) or datasets generated in this study (*S. anglica* and *Solanum lycopersicum*). In these data, we identified homologs of *MMT*, *SDC*, *DOX*, *CAO1* (as well as *EF1A* and *GAPDH* to correct for sequencing depth artefacts) (Supplementary Data 5) and calculated the transcript abundance for each relative to *ACT8* (Fig. 3f). This revealed that *DOX* is much more highly expressed in *Spartina anglica* relative to the other low producing species (33–84-times). *MMT* transcript levels were equivalent between *S. anglica* and the other species, except, unexpectedly, in *H. vulgare*, which demonstrated highly elevated levels of *MMT*. *SDC* levels in *S. anglica* were roughly double those of other species. Overall, these data demonstrate that DOX activity is common in plants, and there is little enzymatic difference between DOX from high producing *S. anglica* and other low producing species. However, in *S. anglica*, *DOX* is much more highly expressed, and therefore DOX activity is substantially higher overall.

### DMSP functions in plant stress and can improve stress tolerance

To determine how DMSP functions in plants, we used tomato as a model, since this species produces DMSP in a salinity- and drought-regulated manner[14] and is highly amenable to laboratory experiments. We subjected plants to salt stress in the presence/absence of DMSP. DMSP was taken up by the roots and transported into aerial tissues, resulting in DMSP accumulation in the leaves 4–6-fold higher than in untreated plants (up to maximum levels of 21 nmol g$^{-1}$ FW; Supplementary Fig. 7a). As expected, salt stress resulted in significantly decreased biomass (42% lower than control plants; Fig. 4a, b). Strikingly, addition of DMSP resulted in total biomass rescue (Fig. 4a, b), demonstrating the protective effects of this compound in plants.

To dissect further the mechanisms by which DMSP acts in planta, we performed RNA-seq on plants grown in the presence of salt and/or DMSP (Supplementary Fig. 7b). Salt stress resulted in large-scale transcriptomic changes relative to control plants (Supplementary Fig. 7c), with 1033 transcripts differentially expressed by 2-fold or greater in NaCl treated plants compared with control plants. In NaCl and DMSP treated plants, this number was decreased to 704 transcripts, of which 427 were also differentially expressed between NaCl treated plants and the untreated control. Gene ontology enrichment on genes that were differentially expressed in response to DMSP treatment alone identified enrichment in processes relating to water movement and auxin signalling (Supplementary Fig. 7d), suggesting association with salt stress responses[38]. A total of 175 genes were differentially expressed in plants treated with NaCl and DMSP (Fig. 4c) relative to NaCl treated plants. Gene ontology enrichment revealed that many of these genes were involved in developmental processes (Supplementary Data 6), as expected given NaCl treated plants were developmentally arrested

whilst NaCl and DMSP treated plants were not (Fig. 4a, b). Of these genes, many of the most differentially expressed (or their homologs in other species) have roles in osmotic stress (*SlSWEET*, *SlACO*, *SlFIBRILLIN*, *SlTIP3;2*)[39–42], oxidative stress (*SlGLO2*, *SlFER2*, *SlOPT3*, *SlLEA5*, *SlERF105*)[43–47] or salt/metal stress (*SlTDT*, *SlPME13*, *SlPCR2*, *SlREC2*, *SlSTZ*)[48–52] (Fig. 4d). These genes may therefore represent pathways through which DMSP exerts its protective effect. Taken together, these data demonstrate that DMSP can ameliorate salt stress, and likely does so in tomato by counteracting osmotic stress and oxidative stress. This aligns well with our *S. anglica* RNA-seq (Fig. 2c), suggesting DMSP likely acts in a conserved way across species.

Finally, we set out to demonstrate that DMSP synthesis is a trait which can be engineered in plants. Transient expression of *SaMMT1*, *SaSDC* and *SaDOX* alone and in combination in *N. benthamiana* demonstrated that over-expression of all three genes was required to elevate DMSP concentrations significantly (Fig. 4e). Importantly, these transiently transformed plants with elevated DMSP accumulation (23 nmol g$^{-1}$ FW) demonstrated drought resistance relative to control plants (Supplementary Fig. 7e). Transgenic *A. thaliana* plants over-expressing *SaMMT1*, *SaSDC* and *SaDOX* were also generated, and the resulting lines accumulated DMSP up to 3.4 μmol g$^{-1}$ FW (Fig. 4f), i.e. concentrations of DMSP equivalent to *S. anglica* (Fig. 2b). Seedlings of transgenic lines 1, 2 and 4 (3.4, 1.3 and 0.2 μmol g$^{-1}$ FW, respectively) and wildtype Col-0 (10 nmol g$^{-1}$ FW) were grown and subjected to NaCl stress. These transgenic plants did not demonstrate any growth defects, and, excitingly, when exposed to 100 mM NaCl, demonstrated a significant increase in root biomass relative to control plants (Fig. 4g, h), similar to what was seen through root uptake of DMSP (Fig. 4a). Lines 2 and 4 also demonstrated increased aerial biomass relative to the control. Interestingly, the highest accumulating line (line 1) showed no significant difference in shoot biomass, indicating that there is a point above which the protective effects may be lost. This may be a consequence of the DMSP concentrations or of the metabolic cost to synthesise so much. Overall, these data demonstrate that DMSP levels can be manipulated in low DMSP-accumulating species through root uptake of DMSP or by over-expression of *MMT*, *SDC*, and *DOX* from *S. anglica*, and that doing so can increase tolerance to salt stress and drought.

## Discussion

In saltmarshes, cycling of *Spartina* derived DMSP is considered to be the key source of climate active DMS[10]. All plants tested both here and in other studies produce DMSP (Fig. 3a[14,15]), though many do so at low concentrations. Many of these low DMSP-accumulating species are crop plants that cover large areas; for example, in 2022 in the UK, wheat and barley covered 1,809,000 and 1,104,000 hectares, respectively[53]. Cumulatively, terrestrial environments may therefore play host to significant DMSP cycling, a characteristic previously thought to be confined exclusively to marine environments. Indeed, one would predict that environments associated with plants that accumulate high DMSP concentrations contain higher proportions of microorganisms able to catabolise DMSP, and thus produce more DMS, than those from plants accumulating low DMSP concentrations. To test this hypothesis, we analysed a recently published metagenomic dataset of sediments associated with *S. alterniflora* (high DMSP producer) and *S. patens* (low DMSP producer[54]). Although there were some differences in the abundance of specific DMSP lyase genes, notably in *dddY* (Supplementary Data 7), surprisingly, there was no significant difference between the high and low DMSP accumulators in the cumulative abundance of known DMSP lyase genes (15.9 ± 2.4 vs 20.5 ± 3.7%) or the DMSP demethylation gene (8.3 ± 1.7 vs 8.6 ± 1.3%) (Supplementary Fig. 8). The reason for this could be because: the transcript and protein abundance for these genes and the DMSP catabolic rates (not examined in ref. 54)

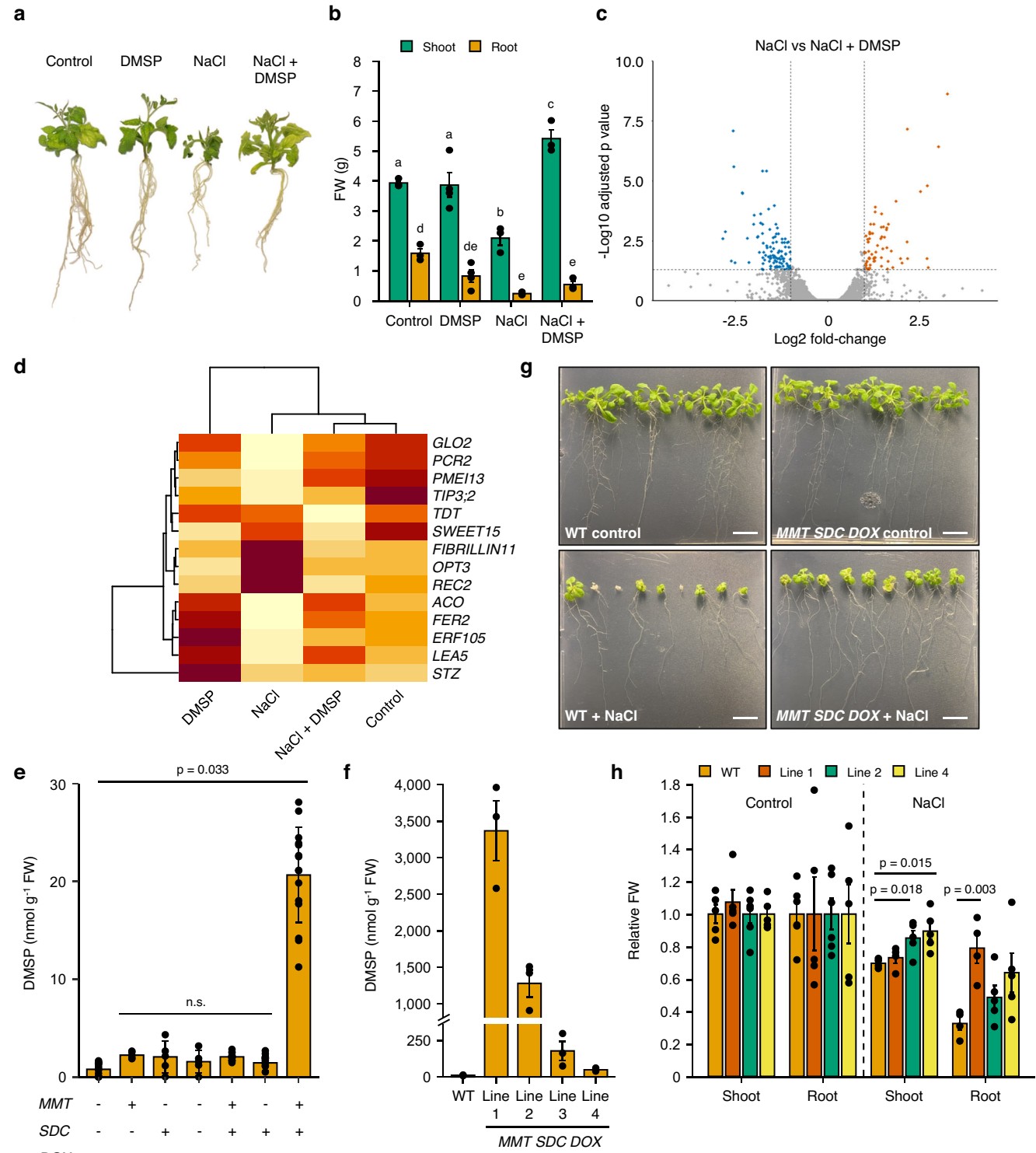

differed between these samples; *S. alterniflora* retains DMSP for its important physiological roles; or other organisms (e.g. bacteria) provide the DMSP to this environment (as implied by ref. 55). Importantly, these terrestrial DMSP lyase gene abundances are very similar to those previously reported for the Earth's surface oceans[56] and coastal surface marine sediments[55]. The significance of terrestrial DMSP cycling thus demands further investigation to determine the significance of these processes in diverse environments, particularly in the context of other microbial activities, e.g. carbon fixation and nitrogen fixation as recently identified in the microbiome of *S. alterniflora*[57].

Our identification and characterisation of the genes presented here constitute the first molecular genetic study of DMSP synthesis in plants. *MMT* and *ALDH* (Fig. 1a) are ubiquitous in plants[17,20] regardless of how much DMSP they produce, and as such are not clear determinants of high-level DMSP production. MMT enzyme activity and gene expression is equivalent between low and high DMSP producing plants (Supplementary Fig. 1a; Fig. 3f). *DOX* genes also appeared near-ubiquitous in plants, showed little variation (Fig. 3d), and were enzymatically similar between species (Fig. 3e). However, in *S. anglica*, this gene is subject to elevated expression (Fig. 3f), resulting in overall much greater DOX activity (Supplementary Fig. 6c). Importantly, our

**Fig. 4 | DMSP protects plants from abiotic stress. a** Image of representative tomato plants from each treatment group, demonstrating that biomass loss caused by NaCl is rescued by DMSP treatment. **b** Fresh weight measurements of tomato plants from each treatment group, demonstrating that biomass loss caused by NaCl is rescued by DMSP treatment ($n = 9$, $p < 0.05$ in one-way ANOVA followed by Tukey tests for shoot and root, separately). **c** Volcano plot demonstrating contrasting gene expression profiles of tomato plants treated with salt stress alone (NaCl) or salt stress and DMSP (NaCl + DMSP). Each dot represents a gene, with red dots indicating genes that are more highly expressed in NaCl treatment, blue dots representing downregulation, and grey dots denoting genes that are not significantly differentially expressed. Dotted lines indicate significance thresholds from DESeq2 analysis. **d** Clustered heatmap of average normalised count numbers of differentially expressed genes of interest from panel c, showing that samples from Control and NaCl + DMSP-treated plants cluster together away from samples from NaCl-treated plants. **e** DMSP measurements of *Nicotiana benthamiana* leaves transformed with different *S. anglica* DMSP synthesis genes, showing that all three genes (*MMT*, *SDC* and *DOX*) are needed for high DMSP accumulation ($n = 15$ for negative control, *MMT SDC*, *SDC DOX*, and *MMT SDC DOX*, $n = 5$ for *MMT*, *SDC*, *DOX*). *p*-value indicates statistical significance after one-way ANOVA followed by Tukey test; n.s. denotes no statistically significant difference. **f** DMSP measurements of *Arabidopsis thaliana* over-expressing *S. anglica MMT*, *SDC* and *DOX* showing highly elevated accumulation in four independent transgenic lines ($n = 3 \times 18$ plants). **g** Images of wildtype (WT, Col-0) and transgenic *A. thaliana* seedlings (Line 2) growing in the presence and absence of NaCl, demonstrating over-expression of *S. anglica* DMSP synthesis genes (*MMT*, *SDC* and *DOX*) convey tolerance to salt stress. Scale bar = 1 cm. **h** Fresh weight measurements of plants in (**g**), demonstrating biomass rescue is statistically significant ($n = 6 \times 9$ plants, total 54 per condition). *p*-values indicate statistical significance after one-way ANOVA followed by Tukey test. Data represent mean ± one standard error. Source data are provided as a Source Data file.

work demonstrates that SDC is the most critical enzyme for evolution of high-level DMSP synthesis in *Spartina*, with the *S. anglica* SDC demonstrating markedly different kinetics with SMM than homologous ODC enzymes from other plant species (Fig. 3d). Indeed, this difference was salient when comparing with the ODC of *S. patens*, a species that diverged fewer than 2–4 MYA from *S. anglica*[58] but that fails to produce high concentrations of DMSP (Fig. 3a). In addition to highlighting the importance of evolution of ODC to SDC for high-level DMSP synthesis in the *Spartina* lineage, this comparison also provides a timeframe in which this change occurred. Additionally, the lack of detectable SDC activity or expression in other high DMSP producers (*Saccharum officinarum*, *P. oceanica*, *Z. marina*) suggests the existence of other plant DMSP synthesis routes that are SDC-independent (e.g. as in *M. biflora*[59]), which warrant future investigation. Overall, we conclude that high-level DMSP synthesis evolved convergently multiple times in plants.

DMSP is an anti-stress molecule and levels of DMSP in *S. anglica* growing in the environment showed substantial variation, which was unlinked to the primary metabolic state of plants (Fig. 2). DMSP concentrations showed a positive correlation with plant molybdenum stores, and a negative correlation with nitrogen and levels of proline (Fig. 2e), a protein-coding amino acid that also serves as an osmolyte and anti-stress molecule in plants[60]. Given the highly saline nature of saltmarshes, *S. anglica* must maintain elevated pools of osmolytes, but this represents a substantial drain on nitrogen reserves, which are frequently limiting in *Spartina* species[61]. Sulfur is rarely limiting in saltmarshes owing to daily fluxes of sulfates from tides[62], so DMSP likely represents a component of a diverse toolkit of osmolytes that *S. anglica* can draw upon to maintain its water potential when met with specific nutrient limitations. Additionally, as ammonia is released from DMSP-amine to produce DMSP-aldehyde (Fig. 1b), this synthesis pathway could ultimately also be used to recycle nitrogen from methionine pools, thereby helping to balance the nitrogen and sulfur demands of the plant.

By modulating DMSP levels through root uptake or over-expression of *MMT*, *SDC* and *DOX*, we demonstrate that DMSP can improve the tolerance of plants to salt stress and drought (Fig. 4). The mechanism of action of DMSP is likely through its capacity to act as an osmolyte and counteract oxidative stress (Fig. 4d). Such an approach may be of particular benefit in nitrogen-poor soils, where production of nitrogen-containing osmolytes would be a significant drain on the limited nitrogen levels of a plant. Manipulation of DMSP levels in plants through root uptake and the use of transgenics to engineer DMSP production represent potential routes to enhance abiotic stress resistance and improve agricultural productivity. Whether transgenic plants with elevated levels of DMSP would also significantly alter DMS flux into the atmosphere remains to be explored, but this would be an exciting area of future study, particularly in the face of global climate change.

## Methods

### Spartina anglica sampling
*S. anglica* leaf tissue was collected from Stiffkey saltmarsh in Norfolk, UK (Supplementary Fig. 4a). Tissue was washed three times using deionised water to remove surface contaminating bacteria and then RNA and protein was extracted immediately for transcriptomics and protein activity assays. For each plant, soil was also collected from the base of the plant for ICP-AES and CHN analyses.

### RNA extraction from *Spartina anglica* and assembly of transcriptome
For each *S. anglica* sample, leaves were collected from 4–6 clumped individuals. These were then cut into segments, pooled, and 1 g was homogenised in liquid nitrogen. RNA was extracted using Tri Reagent Solution (ThermoFisher Scientific, AM9738) as per the manufacturer's instructions. RNA was cleaned up using Zymo RNA Clean & Concentrator™-25 (Zymo-Research). Extracted RNA was then sent to Novogene Co. where mRNA libraries were prepared. The mRNA libraries were sequenced with paired-ends at a read depth of approximately 20 million reads. Adapters were trimmed from reads using Trim Galore[63] and a reference transcriptome was built using Trinity[64] under default settings. For highest and lowest DMSP accumulators, a reference transcriptome was constructed using both sets of samples, and then redundancy was removed.

### Identification of candidate genes from *Spartina anglica* transcriptome
For each gene of interest (*MMT*, *SDC*, *DOX* and *ALDH*), a list of protein sequences containing the domain of interest was curated for *A. thaliana*, *Z. mays* and *T. aestivum*, using the InterPro entries IPR025779, IPR008286, IPR000269 and IPR015590, respectively. A tBLASTn search was run against the *S. anglica* transcriptome using each of the lists as query sequences. The contigs identified by this BLAST search were then subset so that only those with an e-value lower than 0.0005 were carried forward. For *MMT*, *SDC* and *DOX* lists, the non-redundant contig sequences were then manually translated and continuous reading frames with start and stop codons were extracted. For each open reading frame (ORF), a BLASTp search was run against the complete BLAST database, and only those which had near complete alignment (>90%) to matching genes of interest were carried forward. The ORFs were aligned against one another and redundant ORFs were discarded. Finally, the nucleotide sequences for each of the remaining ORFs were identified from the original contigs, and these were domesticated for Golden Gate cloning and synthesised via commercial DNA synthesis. This resulted in two transcripts for *MMT*, three for *SDC* and two for *DOX* (Supplementary Data 2).

### Protein extractions from *Spartina anglica*
For DOX activity tracking, 60 g of *S. anglica* leaf tissue was used for protein extraction. The plant material was blended in 150 mL buffer

containing potassium phosphate buffer 50 mM pH 8, DTT 5 mM, EDTA 1 mM, L-ascorbic acid 5 mM (Buffer A), for purification of DOX activity[17]. Blended samples were filtered through three layers of Miracloth (Millipore), followed by dialysis with SpectraPor™ dry standard grade dialysis tubing MWCO 6–8000 (Repligen) overnight. Dialysis was performed in 5 litres of Tris 20 mM pH 8, NaCl 150 mM, which was changed three times. Samples were then centrifuged for 45 min at 4000 g, and the supernatant was taken forward. Protein was then quantified by Bradford assay and brought to a final concentration of 2 mg mL$^{-1}$. All steps were performed at 4 °C.

For all other enzyme extract experiments, 5–10 g of leaf tissue was used, and homogenisation was performed under liquid nitrogen using a mortar and pestle. For SDC assays, powder was thawed in potassium phosphate buffer 50 mM pH 7.2, DTT 5 mM, EDTA 1 mM, pyridoxal 5′-phosphate 0.1 mM, L-ascorbic acid 5 mM (Buffer B), centrifuged at 4000 g for 10 min to clarify, and then filtered through Miracloth. Material for DOX assays was treated the same, but thawed in Buffer A. Filtered extracts were then desalted by PD-10 columns (Cytiva) using gravity filtration and exchanged into fresh Buffer B for SDC, and Buffer A with no L-ascorbic acid for DOX.

### Isolation of DOX by activity tracking

Protein extract was brought to 40% ammonium sulfate over 1 h and was incubated at 4 °C for 1 h. The protein extract was centrifuged for 30 min at 4000 g, and the supernatant was dialysed overnight using SpectraPor™ dry standard grade dialysis tubing MWCO 6–8000 (Repligen). Proteins were then fractionated on a phenyl sepharose column (Cytiva) with a programme of 0–100% PS Buffer A (Tris-HCl 50 mM pH 8, 20% Ammonium sulfate) against PS Buffer B (Tris-HCl 50 mM pH 8) over 400 mL. Two fractions demonstrated activity and the fraction with the highest activity was taken forward for Q anion exchange chromatography, in which the proteins were fractionated across a gradient of Q Buffer A (Tris-HCl 50 mM pH 8, NaCl 1 M) 0–100% against Q Buffer B (Tris-HCl 50 mM pH 8, NaCl 50 mM) over 400 mL. A single fraction demonstrated DOX activity. This final fraction was separated by gel filtration with Superdex HiLoad 16/600 Column (Cytiva) with an isocratic flow of Q Buffer B. After each fractionation, DOX activity demonstrated an enrichment (Supplementary Fig. 3b). Two samples were sent for LC-MS analysis (Supplementary Fig. 3b) at the John Innes Centre Proteomic Platform. These samples were the final active fraction after gel filtration chromatography and the lower activity fraction taken from the phenyl sepharose chromatography.

### Purification of recombinant proteins from *E. coli*

Genes of interest were cloned into a pMALc2x plasmid modified for Golden Gate cloning and then transformed into *E. coli* strain BL21 DE3. Positive colonies (verified by Sanger sequencing) were grown overnight at 37 °C 180 rpm in 10 mL Luria-Bertani (LB) broth containing 100 mg mL$^{-1}$ carbenicillin. These 10 mL cultures were then used to inoculate 1 litre cultures (LB broth containing 100 mg mL$^{-1}$ carbenicillin and 0.05% glucose), which were grown at 37 °C 180 rpm to an OD$_{600}$ of 0.6 and then induced with 0.5 mM IPTG at 28 °C 180 rpm for 3 h. Cultures were then pelleted, resuspended in lysis buffer (Tris-HCl 20 mM pH7.4, NaCl 200 mM, EDTA 1 mM, DTT 1 mM) and lysed by French press at 10000 psi. Samples were separated into soluble and insoluble fractions by centrifugation at 9000 g for 30 min, and the soluble fraction was then incubated for 1 h with amylose resin (New England Biolabs E8021S) pre-washed with washing buffer (Tris-HCl 20 mM pH 7.4, NaCl 200 mM, EDTA 1 mM) four times. We confirmed that each protein of interest was in the soluble fraction by running samples of total lysed extract, soluble fraction, and insoluble fraction on polyacrylamide gels. After incubation with amylose resin, samples were centrifuged for 1000 g for 1 min and the supernatant removed. The resin was washed four times with washing buffer, and then poured into an empty PD-10

column with filter (Cytiva). Wash buffer was allowed to run through by gravity, followed by a wash with washing buffer 2 (Tris-HCl 20 mM pH 7.4, NaCl 200 mM). Columns were then incubated for 5 min with 10 mM maltose (Merck 63418) dissolved in washing buffer 2, and the eluate was collected and quantified by Bradford assay.

### Enzyme activity assays

For total plant extracts 20–40 µg of protein was used, and for purified enzymes 2–10 µg of protein. All enzyme reactions were performed at 25 °C. To assay MMT activity, protein extracts or purified enzymes were incubated with 1 mM L-methionine (Merck 1.05707) and 1 mM *S*-adenosylmethionine (Merck 798231) in a reaction buffer containing Tris-HCl 20 mM pH 7.2 and DTT 1 mM.

Decarboxylase assays were incubated with 1 mM substrate (L-*S*-methylmethionine (Biosynth FM25160), L-ornithine (Merck O2375), L-diaminopimelic acid (Merck 33240) or L-arginine (Merck A5006)) in a buffer containing Tris-HCl 20 mM pH 7.2 and pyridoxal 5′-phosphate 0.1 mM (Merck P9255). For plant extracts, 5 mM L-ascorbic acid was also included in this mixture to inhibit removal of the product by DOX activity. Decarboxylases were measured by HPLC and NMR for the kinetics. Note: for experiments with SaSDC, 10 µg of protein was used for SDC assays, but 35 µg was used for ornithine assays, owing to a lack of detectable activity.

DOX assays were incubated with 1 mM substrate (unless otherwise specified), either putrescine (Merck 51799) or DMSP-amine which was synthesised enzymatically using our identified SDC and confirmed by HPLC. Reactions were set up in Tris-HCl 20 mM pH 8 in crimped 2 mL gas-tight glass vials and incubated at 25 °C in the dark. As DMSP-aldehyde is unstable, it decays with a half-life of ~1 h[17], resulting in DMS in the headspace of these vials. Measurements by gas chromatography were then taken of headspace DMS from decayed DMSP-aldehyde after a minimum of 2 h.

### High performance liquid chromatography

Enzyme assays were precipitated by the addition of 100% tri-chloroacetic acid (Merck T0699) to a final concentration of 0.15%. Samples were incubated at 4 °C for 10 min and then centrifuged at 13000 g for 10 min. The supernatant was transferred to a new tube and 50 µL of this supernatant was then reacted with an equal volume of ortho-phthaldialdehyde (Merck P0532) for 5 min for the derivatisation reaction to occur. The fluorescent adducts were then resolved via reverse-phase chromatography by injecting 20 µL of sample on to a Synergi™ 4 µm Hydro-RP 80 Å 100x2 mm column (Phenomenex) using buffers A (NaH$_2$PO$_4$ 25 mM pH 2.5 adjusted with H$_3$PO$_4$) and B (50:40:10 methanol:acetonitrile:water)[65], with a flow rate of 0.3 mL min$^{-1}$. For our method, we ran a gradient of 90% A and 10% B to 20% A and 80% B over 12 min, which was then held for 5 min. The column was then brought back to 90% A and 10% B over 2 min and allowed to equilibrate for a further 5 min. A Dionex RF2000 fluorescence detector (Dionex) was used to detect the products with excitation at 332 nm and emission at 445 nm.

For sulfate measurements, 10 mg of dry ground sample was extracted in 200 µL water for 30 min with sonication in a bath sonicator. The extract was diluted 100-fold in water, and 10 µL was injected onto a Dionex Ion Pac AS18 column 2 x 250 mm, fitted with AG18 2 x 50 mm guard column. The column was eluted with a gradient of potassium hydroxide delivered at a flow rate of 0.25 mL min$^{-1}$ by a Dionex ICS-2100 Ion Chromatography System and Dionex AS-AP autosampler. The gradient was delivered with the schedule: time (min), mM; 0,12; 12,12; 20,34; 25,100; 25.1,12; 35,12. Sulfate was detected as a peak with retention time of 12.18 ± 0.12 min. Calibration curves for injection of 10 pmol to 10 nmol gave $R^2 > 0.998$ and peaks were integrated in Dionex Chromeleon v.6 software.

For *O*-acetylserine measurements, 10 mg of dry ground sample was extracted in 200 µL water for 30 min with sonication in a bath

sonicator. The extract derivatised with ortho-phthaldialdehyde (Merck P0532), and samples were resolved as described for enzyme assay measurements with the following differences: 10 μL of sample was injected onto the column, and the gradient was delivered with the schedule: time (min), % B: 0,20; 20,80; 25,80; 26,20; 31,20. The retention time of O-acetylserine was 8.11 ± 0.05 min. Calibration curve for injection of 25–500 pmol gave $R^2 = 0.998$, and peaks were integrated in Jasco ChromNav v.1 software.

## Gas chromatography

DMSP measurements were conducted using gas chromatography (GC). For plant samples, a known quantity of tissue (usually between 30–50 mg) was placed in a 2 mL glass vial containing 0.1 mL deionised water. The sample was homogenised using a pipette tip, 0.1 mL of 10 M NaOH was added to each homogenate to liberate DMS from DMSP by alkaline lysis and followed by immediate crimping. Samples were incubated in the dark at room temperature overnight and assayed the subsequent day. DMS in the headspace of these vials was resolved using an HP-INNOWax 30 m × 0.530 mm capillary column and measured using a flame photometric detector (Agilent 7890 A GC system fitted with a 7693 autosampler). For all species, the presence of DMSP was confirmed by LC-MS[66].

## Nuclear magnetic resonance

All nuclear magnetic resonance (NMR) measurements of enzyme reactions were performed in a 5 mm NMR tube at 298 K on a Bruker 500 MHz spectrometer with autosampler. Product formation was quantified against an internal standard of 1 mM pyrazine (Merck P56003). The pulse sequence used incorporated a double echo excitation sculpting component for water suppression (Bruker library zgesgp) to remove residual water. Samples were run using 256 scans with a relaxation delay D1 of 1 s. All spectra were phased, base corrected and calibrated for the pyrazine peak at 8.63975 ppm. The chemical shift of the diagnostic groups of SMM and DMSP-amine were the $CH_2$-group adjacent to the sulphur (peak 3 in Supplementary Fig. 2f). The initial rates obtained were then plotted against $[S]_0$ using QtiPlot to obtain the enzymatic parameters $K_m$ and $V_{max}$ as shown in Supplementary Fig. 2e.

## Sequence identification of homologs in other species

The S. anglica MMT, SDC and DOX protein sequences were used in a tBLASTn search against the selected species. For each species, the top hit was taken, then codon optimised for expression in E. coli. Sequences were then synthesised by commercial DNA synthesis and cloned into the Golden Gate modified pMALc2x plasmid. For SpSDC, which lacks genetic resources, DNA was extracted from S. patens material from Foz de Odeleite, Portugal. Primers designed against the full-length CDS of SaSDC (Supplementary Data 8) were used to amplify the S. patens DNA by polymerase chain reaction (PCR) using high fidelity Phusion® polymerase (New England Biolabs M0530). Bands of equivalent size to S. anglica SDC were extracted and cloned into the pJET1.2/blunt cloning vector (Thermo Scientific K1231). E.coli strain DH5α was transformed with this construct, and clones were selected and sequenced by Sanger sequencing using forward and reverse pJet1.2 sequencing primers. Sequences were then aligned against SaSDC and a full-length homolog was identified. This S. patens SDC homolog was cloned into the Golden Gate modified pMALc2x plasmid, as with other homologs.

## RT-qPCR

RNA from S. anglica was extracted as described above. RNA samples were treated with TURBO™ DNAse (ThermoFisher Scientific, AM2238) according to the manufacturer's instructions. Samples were then tested for contaminating DNA by PCR using RT-qPCR SaActin primers (Supplementary Data 8). Complementary DNA (cDNA) first strand synthesis was performed using SuperScript IV® Reverse transcriptase

(Thermo Fisher) according to manufacture protocol. Housekeeper genes SaActin and SaGAPDH were identified in the S. anglica transcriptome by BLAST. Specific primers pairs were designed using NCBI Primer BLAST specifying PCR product sizes of 120–200 bp and annealing temperatures of 60 °C. RT-qPCR used total leaf RNA extracted from three independent biological replicates and was performed with three technical replicates, in 96-well plates using the RealTime PCR System (Aria system, Aglient). Primers showing 90–110% efficiency were used (Supplementary Data 8) and amplification was performed using a programme consisting of 94 °C for 2 min, followed by 40 cycles of 94 °C for 15 s and 60 °C for 1 min, using SYBR® Green JumpStart™ Taq ReadyMix™ (Merck S5193) according to the manufacturer's protocol. For analysis, the Ct value of the target gene was normalised and expression values relative to each control were then determined by the Pfaffl method[67].

## Quantification of amino acids by liquid chromatography-mass spectrometry

Leaf tissue from each of the three clumps of S. anglica from each of the five sampling sites was homogenised under liquid nitrogen and freeze dried. Amino acids were extracted from 30 mg of this powder. Extracted samples were then derivatised with AccQ Tag (Waters) and diluted 1:10–1:1000 to bring in line with standard curves. Measurements were then performed by liquid chromatography-mass spectrometry (LC-MS)[68].

## Differential expression analyses

Tomato and S. anglica mRNA libraries were constructed and sequenced with paired-ends by Novogene Co. Adapters were removed from reads using Trim Galore[63]. The tomato experiments were performed in quadruplet, and two of these biological replicates were then taken forward for RNA sequencing. For the S. anglica experiments, material from three independent biological replicates was used for each group (i.e. the three plants with the highest or lowest DMSP accumulation). Clean reads were then aligned either to the reference tomato transcriptome iTAG v4.1 for tomato sequencing or to our own trinity transcriptome constructed from S. anglica RNA-seq using HISAT2[69] and default parameters. Aligned reads were then quantified using Salmon[70] run in alignment-based mode. Statistical testing of quantified reads was then performed using DESeq2[71]. All gene ontology enrichments were performed using ShinyGO v0.77[72].

## Inductively coupled plasma atomic emission spectroscopy

Plant samples were homogenised in liquid nitrogen, freeze dried and then digested in nitric acid. Soil samples were freeze dried and 1 mL volume of powder was incubated in 5 mL of 50 mM ammonium EDTA (Merck 330507) with shaking for 1 h at room temperature. Samples were then centrifuged at 4000 g for 10 min and filtered through a 0.22 μm filter. Plant digests and soil solutions were then analysed for elemental composition using an Agilent Vista Pro at the UEA Science Analytical Facility.

## Nitrogen measurements

Plant and soil samples were homogenised at room temperature and freeze dried. 1.5 mg material was then weighed into nickel sleeves and assayed using Exeter CE440 CHN analyser calibrated using acetanilide and benzoic acid standards.

## Construction of phylogenetic trees

BLASTp was run using SaMMT1, SaSDC and SaDOX as input sequences. The top 50 hits were taken along with homologous sequences from S. officinarum, S. lycopersicum and M. truncatula. SaCAO1 and SaDAPDC were included to root the DOX and SDC trees, respectively. For SaMMT, trees were rooted against mmtN from Novosphingobium[55]. Rooting branches were then removed for visual clarity. Sequences

were aligned using MUSCLE[73] and the subsequent Phylip file was run through PhyML[74] with a standard bootstrap analysis of 100 repeats. The most similar hit for each species was retained and lower similarity accessions were removed so that each species was represented only once on each tree.

## Quantification of *MMT, SDC* and *DOX* abundance in RNA-seq datasets

Primary RNA-seq data were collected for *S. anglica*, *S. lycopersicum*, *A. thaliana*, *H. vulgare* and *N. benthamiana* from the sources indicated in Supplementary Data 5. Adapters were trimmed as before and clean reads were aligned using Salmon[70] to de novo assembled transcriptome for *S. anglica*, iTAG v4.1 for *S. lycopersicum*, TAIR10 cDNA for *A. thaliana*, BaRT2v18 for *H. vulgare* and NBv5 for *N. benthamiana*. Mapping parameters were kept the same for all species. Homologs were identified for genes of interest using BLAST and reads mapping to them were then taken from each of the quantitative alignment files. Finally, for each library, the total number of reads mapping to each gene were divided to the total number of reads mapping to the *ACT8* homolog to remove artefacts that may be introduced due to differences in sequencing depth and transcriptomic complexity.

## Plant materials and growth conditions

Tomato (*S. lycopersicum* cv. Micro Tom) seeds were obtained from Moles Seeds (www.wholesale.molesseeds.co.uk) and sterilised using 70% ethanol for 2 min, followed by 1% sodium hypochlorite (Merck) for 20 min and rinsed five times with sterile water prior to sowing. Tomato growth media was prepared (Murashige and Skoog basal medium (Merck M5519), 3% sucrose, 0.8% agar) and five seeds were sown per 40 mL of media in a Magenta GA-7 Plant Culture Box (Bioworld) using sterile technique. Material was placed in dark conditions at 4 °C for five days before transfer to a Sanyo MLR-351 controlled environmental chamber, at 26 °C/18 °C, day/night with a 16 h photoperiod, for 2 weeks of growth. Plantlets at the 3–4 true leaf stage were transferred to treatment conditions using 40 mL fresh media, three plantlets per GA-7 box. Tomato growth medium was supplemented with ± 5 μM DMSP or 100 mM NaCl ± 5 μM DMSP. Phenotypes were observed after 2 weeks of growth under treatment conditions. Fresh weight was recorded for aerial and roots tissues per box. DMSP levels were measured for leaf. For each condition, four biological replicates of three technical replicates were used. For RNA-seq, two boxes were selected per condition.

For *A. thaliana* experiments, seedlings were sterilised in 1% sodium hypochlorite for 13 min and then rinsed five times with sterile water. Seedlings were sown on 1/2 strength MS containing 0.25% glucose, stratified at 4 °C for two days, and then grown in a Sanyo MLR-351 controlled environmental chamber, at 22 °C/18 °C, day/night with a 16 h photoperiod, for 12 days of growth. Plants were then transferred to treatment plates containing fresh 1/2 MS with 0.25% glucose ± 100 mM NaCl and grown for a further 12 days before being harvested for further analyses.

For glasshouse experiments, *Spartina anglica* plants were collected from Stiffkey saltmarsh between May and July 2022 and maintained in freshwater until January 2024. Clonal clumps from six plants were each divided into three, and each sub-divided clump was then treated with either deionised water, sea salts (35 PSU) or NaCl (500 mM) for up to 5 weeks. The salinity of treatments with sea salts or NaCl was measured by electrical conductivity and maintained at 45–55 mS cm⁻¹ throughout the experiment. Three leaf samples from each clump were pooled together and collected 1, 3 and 5 weeks after treatment.

## Plasmid construction and creation of transgenic plants

Plasmids over-expressing *MMT* (*pOsUBI3-SaMMT*), *SDC* (*p35S-SaSDC*) and/or *DOX* (*pAtUBI10-DOX*) were constructed using Golden Gate cloning, and transformation of *N. benthamiana* using *Agrobacterium tumefaciens* was performed[75]. Transgenic *A. thaliana* plants were created by floral dip. Transgenic plants were selected on media containing kanamycin and all subsequent phenotypic analyses were performed on T3 material.

## Quantification of DMSP lyase and demethylation gene abundance in metagenomic datasets

Raw data for *S. alterniflora* (accession numbers: SRR11828899, SRR11829104 and SRR11829000) and *S. patens* (accession numbers: SRR11829102, SRR11828889 and SRR11828593)[54] were retrieved from the NCBI Sequence Read Archive (SRA) using the SRA toolkit v.2.10.2. Data quality was assessed using FastQC v0.12.1, and as the quality was deemed high, no trimming was necessary. Assembly (MEGAHIT v1.2.9), annotation (Prodigal v2.6.3), and non-redundant protein sequence generation (CD-HIT v4.8.1) were performed. Gene relative abundances were estimated by the "metabat" method in CoverM 0.6.1. Profile Hidden Markov Model (HMM)-based searches were conducted for the nine DMSP lyase genes (*dddD*, *dddK*, *dddL*, *dddP*, *dddQ*, *dddU*, *dddW*, *dddX* and *dddY*) and the DMSP demethylation gene (*dmdA*) using HMMER v.3.3 with an e-value threshold of < 1e-30. Sequences were further refined by BLASTp; only homologs with ≥ 40% amino acid identity and ≥ 70% subject coverage to ratified sequences were counted. Relative abundances of *ddd* and *dmdA* genes were normalised to the average relative abundance of ten conserved single copy marker genes (retrieved with an HMM e-value threshold of <1e-10)[76,77].

## Reporting summary

Further information on research design is available in the Nature Portfolio Reporting Summary linked to this article.

## Data availability

RNA-seq data generated in this study have been deposited in the NCBI SRA database under accession codes SAMN42957141 and SAMN42997936. All materials used in this study are available from the corresponding authors upon request. Source data are provided with this paper.

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

## Acknowledgements

We thank Jesús Manuel Castillo Segura for the provision of *Spartina patens* tissue, and Leanne Sims and Simone Payne for assistance with DOX activity tracking. The proteomics analysis was performed by Carlo Martins and Gerhard Saalbach at the Proteomics Facility of the John Innes Centre, Norwich, UK, supported by the BBSRC core capability grant. The amino acid LC-MS measurements were performed by Paul Brett and Baldeep Kular at the Metabolomics Facility of the John Innes Centre, Norwich, UK, supported by the BBSRC core capability grant. Elemental analyses were performed by Graham Chilvers and Antony Hinchliffe at the University of East Anglia Science Analytical Facility, Norwich, UK. We thank Libby Hanwell and Ines Raycheva for assistance with plant sampling, Pam Wells for caring for our glasshouse-grown plants, Hayley Whitfield for assistance with plant sample preparation for HPLC, and the National Trust for allowing us to sample *S. anglica* from Stiffkey saltmarsh. We also thank Tracey Chapman, Tamas Dalmay and Tony Miller for comments on the manuscript. RDP, LB, AJD, CAB, JDT and JBM were funded by the Natural Environment Research Council (NERC; NE/V000756/1). RDP, LB, JDT and JBM were also funded by the Biotechnology and Biological Sciences Research Council (BB/X005968/1). Work in JDT's laboratory was additionally funded by NERC (NE/P012671, NE/S001352, NE/X000990 and NE/X014428) and the Leverhulme Trust (RPG-2020-413). SMTK was funded by a PhD studentship from NERC via the ARIES Doctoral Training Partnership (NE/S007334/1). SM performed this work whilst working with Matthew Wallace, with funding from UK Research and Innovation via a Future Leaders Fellowship to MW (MR/T044020/1). ARN and MGEA were funded by PhD studentships from BBSRC via the Norwich Research Park Doctoral Training Partnership (BB/T008717/1).

## Author contributions

RDP, JDT and JBM conceived and designed the experiments. RDP and LJB performed most of the experiments, and JDT and JBM supervised the project. SMTK constructed de novo transcriptomes of *S. anglica*, identified and analysed expression of *S. anglica* sulfate transporters in the transcriptomic dataset, and performed experiments with glasshouse-grown *S. anglica* plants. SM and ARN performed the NMR work. AJD oversaw environmental sampling work and identified species. MGEA, LJB and RDP collected and assayed plant material for DMSP accumulation. CAB established HPLC methods, and performed sulfate and *O*-acetylserine measurements. SMTK and XYZ analysed the metagenomic dataset. RDP, JDT and JBM interpreted the data and wrote the paper. All authors reviewed and approved the manuscript.

## Competing interests

The authors declare no competing interests.
