## [Peer Review File · Nature Communications]

Elucidation of *Spartina* dimethylsulfoniopropionate synthesis genes enables engineering of stress tolerant plantsReviewers' Comments:

Reviewer #1:

Remarks to the Author:

The manuscript of Payet et al. addresses dimethylsulfoniopropionate (DMSP) synthesis in plants. DMSP is an important component of global sulfur cycle, primarily due to its high production in marine phytoplankton and bacterial degradation to volatile dimethylsulfide (DMS). It has long been known that some plants accumulate DMSP, using a synthesis different from marine algae. The work described in this manuscript provides much deeper characterization of the pathway in one of the major DMSP producers, marsh plant *Spartina anglica*. The authors cloned genes for the three enzymes of DMSP synthesis and characterised the corresponding recombinant proteins. They compared transcriptomes, ionomes, and partial metabolomes of different populations of *S. anglica* differing in DMSP concentrations. This work led to the conclusion that the DMSP synthesis is dynamic, controlled in a complex way with some contribution of post-transcriptional regulation, that DMSP is translocated within the plant and that its level is associated with nitrogen availability and stress.

The authors then show that not all DMSP producing plants use the same pathway as SDC seemed to be limited to *S. anglica*. They further reconstituted DMSP synthesis in tobacco and *Arabidopsis* and showed that it protects against salt stress, as does exogenous DMSP. They thus conclude that DMSP engineering may improve plant stress tolerance.

This work closes an important gap in our knowledge about plant sulfur metabolism and adds new facets to understanding salt tolerance. It highlights the variability of plant metabolism, as even within a few high DMSP producers the pathways are not identical. It will open new questions on the ecological drivers triggering DMSP synthesis and the impact of the pathway on plant biotic interactions, which is a topic left out of the work.

The experiments are well planned and executed, the conclusions well justified. The manuscript is very well written, on a level that should be understandable also to non-specialists and the figures are very clear and of good quality. The supplementary data extend the main manuscript body well except that the gene lists would be much more useful if at least some annotation would have been added.

I have a few comments, ideas for ways to increase the impact of the manuscript:

The work on *Spartina* uses plant material from naturally grown plants, it would be important to test the production also in controlled conditions, test to what extent salt induces its synthesis but mainly, to what extent DMSP is linked to NaCl or sea salt, which contains also high concentration of sulfate.

The link of DMSP to sulfur also needs to be elaborated. For example, Mo accumulation is often linked to S limitation, because molybdate is taken up by sulfate transporters. Is there a link between higher DMSP synthesis and sulfate uptake? Are sulfate transporters induced in *Spartina* roots? Is sulfate concentration altered? DMSP synthesis was also discussed as a way of dissipation of energy by sulfate reduction. DMSP would be a significant sulfur pool in the producers, so the impact on S homeostasis should be addressed in much greater detail.

The authors mention few times DMS and the marshes as important contributor to DMS emissions. However, DMS is produced primarily by bacteria from the algal/plant DMSP. DMS production is a major reason for interest in DMSP, but I am missing a proper discussion in the manuscript. For

example, it would be interesting to look at the microbiomes in DMSP producing and non-producing plants and compare the abundance of DMSP lyase genes. These genes are mostly described in marine bacteria, what is their frequency in soil bacteria? Even if not addressed experimentally, a discussion of the relations between DMSP producers and bacteria producing DMS should be added.

The description of the transcriptomics experiments needs to be improved, I did not find information about number or nature of the biological replicates.

Reviewer #2:

Remarks to the Author:

The manuscript by Payet et al. describes the isolation and characterization of genes involved in the biosynthesis of the organosulfur compound dimethylsulfoniopropionate (DMSP) in cordgrass, *Spartina anglica*. This invasive species grows in salt marshes. High levels of DMSP may protect this plant against salt stress. Three genes were isolated, coding for methionine S-methyltransferase, S-methylmethionine decarboxylase and DMSP-amine oxidase. The S-methylmethionine decarboxylase was found to be similar to ornithine decarboxylase, with a specialized activity with its substrate, even among different *Spartina* species. The activity of the DMSP-amine oxidase is broadly distributed in higher plants, however *S. anglica* achieves high flux through enhanced gene expression. Additional experiments probed the function of DMSP in plant stress. Feeding tomato plants with DMSP had a protective effect against salt stress. Transient co-expression of the three genes in *Nicotiana* leaves protected them against wilting due to drought. Stable transgenic expression in *Arabidopsis* resulted in increased DMSP in tissue, associated with resilience to salt stress, showing the potential usefulness of the genes.

The work is original. Although there had been physiological and biochemical studies cited in the manuscript that identified the enzyme activities that have been studied, to date the genes involved had not been identified. This work is of high significance, given the importance of drought and salt stress in agriculture.

In general, the work supports the conclusions and claims presented. The results in Figure 6a are puzzling though. The graph presents the DMSP content in leaf for four different treatments. The legend states that “the measurements ... demonstrate that DMSP is taken up by the roots and accumulates in leaf tissue”. Yet DMSP content is similar between control, untreated plants and plants treated with DMSP. One would expect the DMSP concentration to be higher in the second group. Next, DMSP concentration was highest in plants treated with NaCl, higher than plants treated with both DMSP and NaCl. How can the protective role of DMSP be explained given these results? In the work with transgenic *Arabidopsis*, the results of physiological experiments are presented with a single line, A, which accumulates DMSP (Figure 4). Was this tested with the other line accumulating DMSP, C, or was the experiment repeated with line A? Supplementary Figure 3 focuses on the confirmation of the identity of the DMSP-amine oxidase by partial purification of the enzyme from plant tissue and mass spectrometry of active fractions. However, the mass spectrometry results are not presented. This is a minor point, but would add relevant information to the manuscript.

Frédéric Marsolais

Reviewer #3:

Remarks to the Author:

In this study, the authors have an interest in the organosulfur compound dimethylsulfoniopropionate (DMSP) as an important component underlying the response of the plant to salinity.

Data is presented showing:

1. Identification and characterization of genes involved in DMSP biosynthesis in *Spartina anglica* -- a cordgrass that produces high levels of DMSP.

The data shown in Figure 1 provides convincing evidence for the presence and activities of Methionine S-methyltransferase (MMT), S-methylmethionine decarboxylase (SDC), and DMSP-amine oxidase (DOX) based on biochemical measurements and assays. In addition, they provide evidence for the expression of the underlying genes in leaves (mostly) in comparison to other tissues.

2. Natural variation in DMSP levels in natural populations of *S. anglica*.

The data presented in Figure 2 compares DMSP levels and global gene expression in “clumps” from different areas of a particular saltmarsh. Gene Ontology analyses are used to correlate DMSP levels with stress-associated genes and correlations are also made with levels of ions, nutrients, and amino acids.

I had difficulty figuring out the rationale for these experiments so, while I was interested in the correlations uncovered, I was not able to conclude anything based on the sampling within the saltmarsh. A better rationale for the experiment (why sampling clumps within one saltmarsh makes sense relative to sampling different saltmarshes) would help as would some link to potential differences in salinity levels in the different regions of the saltmarsh sampled.

3. Evolutionary analysis to shed light on the mechanism underlying *S. anglica*'s ability to produce high levels of DMSP.

The data shown in Figure 3 shows a comparison of DMSP levels and SDC and DOX activity across multiple species. Based on the data, the authors conclude that “biochemical changes in SaODC that alter its function to that of an SDC underlie evolution of high-level DMSP accumulation in *S. anglica*” and that in *S. anglica*, “DOX is much more highly expressed, and therefore DOX activity is substantially higher overall. Thus, their studies suggest biochemical and molecular modifications that may have led to increased DMSP production in *S. anglica*.”

4. Functional analysis of DMSP's role in ameliorating the effects of salinity and drought.

The data presented in Figure 4 represents functional studies to alter levels of DMSP (experimentally and genetically) and determine the effect of the plant's response to abiotic stress. The data with tomato and *Arabidopsis* are very interesting. I did not understand the authors claim that they were showing increased DMSP levels in five transgenic *Arabidopsis* lines expressing the Sa MMT, SDC,

and DOX proteins – I can clearly see two and maybe three. Are the levels in panel f shown relative to wild-type Arabidopsis? A better way to represent the data is needed. In addition, the drought data shown in Supplementary Figure 6 needs some quantification and statistical analysis to demonstrate its significance.

Overall – I felt that this is a very nice paper with some interesting and novel data. It is important to identify determinants of abiotic stress tolerance and to demonstrate their role based on functional assays. While I believe that this paper would be perfectly suited to a discipline-specific journal (e.g., a plant-specific journal like *Plant Cell*, *Plant Physiology*...), I am not convinced that its novelty or potential impact would be of interest to the broader readership of *Nature Communications*.

We thank the three reviewers for their expert opinions on our manuscript. We have taken on board their detailed comments and have fully addressed all of the points raised, as outlined below. We have also corrected some typographical errors.

REVIEWER COMMENTS

Reviewer #1 (Remarks to the Author):

The manuscript of Payet et al. addresses dimethylsulfoniopropionate (DMSP) synthesis in plants. DMSP is an important component of global sulfur cycle, primarily due to its high production in marine phytoplankton and bacterial degradation to volatile dimethylsulfide (DMS). It has long been known that some plants accumulate DMSP, using a synthesis different from marine algae. The work described in this manuscript provides much deeper characterization of the pathway in one of the major DMSP producers, marsh plant *Spartina anglica*. The authors cloned genes for the three enzymes of DMSP synthesis and characterised the corresponding recombinant proteins. They compared transcriptomes, ionomes, and partial metabolomes of different populations of *S. anglica* differing in DMSP concentrations. This work led to the conclusion that the DMSP synthesis is dynamic, controlled in a complex way with some contribution of post-transcriptional regulation, that DMSP is translocated within the plant and that its level is associated with nitrogen availability and stress. The authors then show that not all DMSP producing plants use the same pathway as SDC seemed to be limited to *S. anglica*. They further reconstituted DMSP synthesis in tobacco and *Arabidopsis* and showed that it protects against salt stress, as does exogenous DMSP. They thus conclude that DMSP engineering may improve plant stress tolerance.

This work closes an important gap in our knowledge about plant sulfur metabolism and adds new facets to understanding salt tolerance. It highlights the variability of plant metabolism, as even within a few high DMSP producers the pathways are not identical. It will open new questions on the ecological drivers triggering DMSP synthesis and the impact of the pathway on plant biotic interactions, which is a topic left out of the work.

The experiments are well planned and executed, the conclusions well justified. The manuscript is very well written, on a level that should be understandable also to non-specialists and the figures are very clear and of good quality. The supplementary data extend the main manuscript body well except that the gene lists would be much more useful if at least some annotation would have been added.

We thank the reviewer for their supportive comments. As requested, we have now added annotations to the gene lists in the supplementary data to make them more meaningful (Supplementary Tables 5, 11, 12 and 13).

I have a few comments, ideas for ways to increase the impact of the manuscript:

The work on *Spartina* uses plant material from naturally grown plants, it would be important to test the production also in controlled conditions, test to what extent salt induces its synthesis but mainly, to what extent DMSP is linked to NaCl or sea salt, which contains also high concentration of sulfate. The link of DMSP to sulfur also needs to be elaborated. For example, Mo accumulation is often linked to S limitation, because molybdate is taken up by sulfate transporters. Is there a link between higher DMSP synthesis and sulfate uptake? Are sulfate transporters induced in *Spartina* roots? Is sulfate concentration altered? DMSP synthesis was also discussed as a way of dissipation of energy by sulfate reduction. DMSP would be a significant sulfur pool in the producers, so the impact on S homeostasis should be addressed in much greater detail.

We thank the reviewer for their helpful comments and suggestions on these aspects of our work, which have guided us to conduct new experimentation and make new additions that we feel significantly improve our manuscript, as follows.

We performed the suggested experiments using glasshouse-grown *Spartina anglica* treated with NaCl or sea salts. Plants treated with sea salts (containing sulfate) accumulated significantly higher concentrations of DMSP than plants treated with deionised water (control) or NaCl (Supplementary Fig. 5d), demonstrating that DMSP accumulation is indeed correlated with elevated sulfate concentrations. These results have been included on lines 197-205 and in Supplementary Fig. 5d.

To further elaborate on the link between DMSP and sulfur status, we have assessed expression of sulfate transporters (Supplementary Fig. 5a-c), as suggested, and measured the concentrations of sulfate and the key sulfur status indicator *O*-acetylserine in the highest and lowest DMSP accumulators from the naturally grown plants sampled from the saltmarsh (Supplementary Fig. 4e). We found that expression of some sulfate transporters (including *SULTR2;2* and *SULTR2;6*) was induced in *S. anglica* roots (Supplementary Fig. 5b), suggesting that these transporters may be involved in sulfate (and potentially molybdate) uptake. Analysis of our leaf transcriptomic dataset revealed that those *Spartina* plants that accumulated the highest concentrations of DMSP had higher expression levels of *SULTR3;4*, *SULTR3;6* and *SULTR3;7* in their leaves than those plants that accumulated the lowest concentrations of DMSP (Supplementary Fig. 5c; Supplementary Table S9). Since DMSP concentrations positively correlated with the levels of molybdenum in leaves (Fig. 2e), it is possible that these sulfate transporters contribute to molybdenum accumulation in the leaves. These new data have been incorporated into the manuscript on lines 185-189 and in Supplementary Fig. 5.

In addition, we found no difference in the levels of sulfate or *O*-acetylserine in the highest and lowest DMSP accumulators (new data provided in Supplementary Fig. 4e). There was also no significant difference in cysteine or methionine concentrations between the highest and lowest DMSP accumulators (existing data presented in Fig. 2e). Together, these data demonstrate that DMSP accumulation has no significant overall impact on the sulfur homeostasis of *S. anglica*. These new data are included on lines 206-211 and in Supplementary Fig. 4e.

The authors mention few times DMS and the marshes as important contributor to DMS emissions. However, DMS is produced primarily by bacteria from the algal/plant DMSP. DMS production is a major reason for interest in DMSP, but I am missing a proper discussion in the manuscript. For example, it would be interesting to look at the microbiomes in DMSP producing and non-producing plants and compare the abundance of DMSP lyase genes. These genes are mostly described in marine bacteria, what is their frequency in soil bacteria? Even if not addressed experimentally, a discussion of the relations between DMSP producers and bacteria producing DMS should be added.

Thank you for these comments. Although the focus of our manuscript was solely on the production of DMSP by plants, we agree that it would enhance the significance of our story if we were to study the potential of microbiomes associated with plants that accumulate high DMSP concentrations to catabolise DMSP, and thus produce climate active gases such as dimethylsulfide (DMS) and methanethiol (MeSH). We further agree that it would be insightful to compare this DMSP catabolic potential to that of plants accumulating low levels of DMSP. To answer these specific questions, we analysed recently published metagenomic datasets from the sediments associated with *Spartina alterniflora* (high DMSP accumulator) and *S. patens* (low DMSP accumulator) on which the DMSP catabolic potential had been ignored (Pérez Castro *et al.*, 2023; Appl Environ Microbiol. e00988-23). Although this analysis showed that there were some differences in the abundance of specific DMSP lyase genes (notably in *dddY*) between the high and low DMSP accumulators, there was no significant difference in the cumulative abundance of the *ddd* genes (15.9 ± 2.4 vs $20.5 \pm 3.7\%$) or of *dmdA* (8.3 ± 1.7 vs $8.6 \pm 1.3\%$) (Supplementary Fig. 8; Supplementary Tables 15 and 16). This implies that under the sampling conditions used in Pérez Castro *et al.*, 2023, there was no preference for the presence of bacteria with a greater potential to catabolise DMSP via known pathways in samples from high or low DMSP accumulators. It is possible that we would see a very different scenario were we to study the metatranscriptomes and metaproteomes of these samples (unfortunately, such datasets were not generated in the previously published work though). It is also possible that the DMSP in *Spartina alterniflora* is important to the plant's physiology and is retained, and thus other microorganisms in the sediment might contribute to the DMSP present in the associated sediment, as was implied by Williams *et al.*, 2019; Nature Microbiol. doi: 10.1038/s41564-019-0527-1). The situation may also be different under different stages of the plant's life cycle, for example, during senescence *S. alterniflora* may release DMSP into the environment. These are detailed questions that we will investigate in the future and are in our opinion far outside the scope of this primary study.

The reviewer also asks how the abundance of DMSP catabolic genes differs between these plant sediments and typical marine environments. The *ddd* gene abundance in both the *S. alterniflora* and *S. patens* sediments were very similar to those previously reported for the surface oceans, e.g. Tara Oceans samples (Carrion *et al.*, 2023 Nature Microbiol. doi: 10.1038/s41564-023-01526-4), and in coastal surface marine sediments (Williams *et al.*, 2019; Nature Microbiol. doi: 10.1038/s41564-019-0527-1). We have included this new analysis of published metagenomes in a new Supplementary Fig. 8, Supplementary Tables 15 and 16, and in the discussion text (lines 363-381).

The description of the transcriptomics experiments needs to be improved, I did not find information about number of nature of the biological replicates.

We apologise for this oversight. For clarity, three independent biological replicates were used for our transcriptomics experiments for the highest and lowest DMSP accumulating *Spartina* samples (Supplementary Figure 4b). The number of biological replicates has now been clearly detailed in the Materials and Methods (lines 436-437, lines 640-644).

Reviewer #2 (Remarks to the Author):

The manuscript by Payet *et al.* describes the isolation and characterization of genes involved in the biosynthesis of the organosulfur compound dimethylsulfoniopropionate (DMSP) in cordgrass, *Spartina anglica*. This invasive species grows in salt marshes. High levels of DMSP may protect this plant against salt stress. Three genes were isolated, coding for methionine S-methyltransferase, S-methylmethionine decarboxylase and DMSP-amine oxidase. The S-methylmethionine decarboxylase was found to be similar to ornithine decarboxylase, with a specialized activity with its substrate, even among different *Spartina* species. The activity of the DMSP-amine oxidase is broadly distributed in higher plants, however *S. anglica* achieves high flux through enhanced gene expression. Additional experiments probed the function of DMSP in plant stress. Feeding tomato plants with DMSP had a protective effect against salt stress. Transient co-expression of the three genes in *Nicotiana* leaves protected them against wilting due to drought. Stable transgenic expression in *Arabidopsis* resulted in increased DMSP in tissue, associated with resilience to salt stress, showing the potential usefulness of the genes.

The work is original. Although there had been physiological and biochemical studies cited in the manuscript that identified the enzyme activities that have been studied, to date the genes involved had not been identified. This work is of high significance, given the importance of drought and salt stress in agriculture.

In general, the work supports the conclusions and claims presented. The results in Figure 6a are puzzling though. The graph presents the DMSP content in leaf for four different treatments. The legend states that “the measurements ... demonstrate that DMSP is taken up by the roots and accumulates in leaf tissue”. Yet DMSP content is similar between control, untreated plants and plants treated with DMSP. One would expect the DMSP concentration to be higher in the second group. Next, DMSP concentration was highest in plants treated with NaCl, higher than plants treated with both DMSP and NaCl. How can the protective role of DMSP be explained given these results?

We have realised that there was an error in our labelling within Supplementary Figure 6a, meaning that the “NaCl” and “DMSP” labels were inverted (the labels should be in the same order as in Figure 4b). We apologise for this error and we have corrected the labelling in the revised version of this figure (NB. now Supplementary Figure 7a). The data is now consistent with DMSP being taken up by the roots and accumulating in leaf tissue.

In the work with transgenic *Arabidopsis*, the results of physiological experiments are presented with a single line, A, which accumulates DMSP (Figure 4). Was this tested with the other line accumulating DMSP, C, or was the experiment repeated with line A?

We have now greatly improved our work with the transgenic *Arabidopsis* by repeating the experiment using three independent T3 lines that accumulated high concentrations of DMSP. We have presented this new data in a revised Figure 4 (panel h) and updated the relevant manuscript text (lines 343-351).

Supplementary Figure 3 focuses on the confirmation of the identity of the DMSP-amine oxidase by partial purification of the enzyme from plant tissue and mass spectrometry of active fractions. However, the mass spectrometry results are not presented. This is a minor point, but would add relevant information to the manuscript.

We have now presented the mass spectrometry results as a new table (Supplementary Table 4).

Frédéric Marsolais

Reviewer #3 (Remarks to the Author):

In this study, the authors have an interest in the organosulfur compound dimethylsulfoniopropionate (DMSP) as an important component underlying the response of the plant to salinity.

Data is presented showing:

1. Identification and characterization of genes involved in DMSP biosynthesis in *Spartina anglica* -- a cordgrass that produces high levels of DMSP.

The data shown in Figure 1 provides convincing evidence for the presence and activities of Methionine S-methyltransferase (MMT), S-methylmethionine decarboxylase (SDC), and DMSP-amine oxidase (DOX) based on biochemical measurements and assays. In addition, they provide evidence for the expression of the underlying genes in leaves (mostly) in comparison to other tissues.

2. Natural variation in DMSP levels in natural populations of *S. anglica*.

The data presented in Figure 2 compares DMSP levels and global gene expression in “clumps” from different areas of a particular saltmarsh. Gene Ontology analyses are used to correlate DMSP levels with stress-associated genes and correlations are also made with levels of ions, nutrients, and amino acids.

I had difficulty figuring out the rationale for these experiments so, while I was interested in the correlations uncovered, I was not able to conclude anything based on the sampling within the saltmarsh. A better rationale for the experiment (why sampling clumps within one saltmarsh makes sense relative to sampling different saltmarshes) would help as would some link to potential differences in salinity levels in the different regions of the saltmarsh sampled.

We specifically decided to sample clumps within a single saltmarsh because we were interested to measure the diversity in DMSP accumulation within one saltmarsh and, in doing so, we could focus more on environmental factors that regulate DMSP production, rather than factors due to genetic diversity between *Spartina anglica* populations. We have now significantly revised this section of the manuscript (lines 140-149) to provide a stronger rationale for these experiments.

We did not specifically take salinity measurements at each sample site, but our elemental analysis (which was taken at each sample site) identified no significant correlation between DMSP concentrations in the plants and Na concentrations in the soil (Supplementary Fig. 4d). We have now added a sentence to this effect to the manuscript (lines 204-206). Moreover, in agreement with our environmental sampling, we now include new glasshouse-grown *Spartina* experiments, which show no significant increase in DMSP concentration in plants were watered with NaCl, but a significant increase in DMSP concentrations when plants were watered with sea salts (see Reviewer #1 comments above). These environmental sampling and controlled experiments corroborate each other and support the finding that DMSP production in *S. anglica* is not significantly regulated by NaCl (consistent with Mulholland & Otte, 2002 and Van Diggelen *et al.* 1986), but is more likely regulated by sulfates present in sea salts. We feel that our revisions in response to these comments from Reviewers #1 and #3 have now significantly strengthened this section of the manuscript and we thank them for their helpful suggestions.

3. Evolutionary analysis to shed light on the mechanism underlying *S. anglica*'s ability to produce high levels of DMSP.

The data shown in Figure 3 shows a comparison of DMSP levels and SDC and DOX activity across multiple species. Based on the data, the authors conclude that “biochemical changes in SaODC that alter its function to that of an SDC underlie evolution of high-level DMSP accumulation in *S. anglica*” and that in *S. anglica*, “DOX is much more highly expressed, and therefore DOX activity is substantially higher overall. Thus, their studies suggest biochemical and molecular modifications that may have led to increased DMSP production in *S. anglica*.”

4. Functional analysis of DMSP’s role in ameliorating the effects of salinity and drought.

The data presented in Figure 4 represents functional studies to alter levels of DMSP (experimentally and genetically) and determine the effect of the plant’s response to abiotic stress. The data with tomato and *Arabidopsis* are very interesting. I did not understand the authors claim that they were showing increased DMSP levels in five transgenic *Arabidopsis* lines expressing the Sa MMT, SDC, and DOX proteins – I can clearly see two and maybe three. Are the levels in panel f shown relative to wild-type *Arabidopsis*? A better way to represent the data is needed. In addition, the drought data shown in Supplementary Figure 6 needs some quantification and statistical analysis to demonstrate its significance.

We have improved the data presentation in panel f of Figure 4 by using a broken axis so that data from wildtype and all transgenic lines are now clearly visible. For clarity, we have now only shown the data for the four transgenic lines that accumulate DMSP at statistically significantly higher concentrations than wildtype plants. We have now also repeated the phenotypic characterisation of these transgenic plants using three independent T3 lines (panel h; see comment above from Reviewer #2). For the drought data in Supplementary Fig. 6, we have now also included an appropriate statistical test (chi-squared test) to demonstrate its significance (NB. now Supplementary Fig. 7).

Overall – I felt that this is a very nice paper with some interesting and novel data. It is important to identify determinants of abiotic stress tolerance and to demonstrate their role based on functional assays. While I believe that this paper would be perfectly suited to a discipline-specific journal (e.g., a plant-specific journal like *Plant Cell*, *Plant Physiology*...), I am not convinced that its novelty or potential impact would be of interest to the broader readership of *Nature Communications*.

We are glad the reviewer feels that this is an interesting paper with novel data. We hope the suggestion by Reviewer #1 to include an analysis of metagenomics data from high and low DMSP producers helps to broaden the interest of the manuscript further to other disciplines.

Reviewers' Comments:

Reviewer #1:

Remarks to the Author:

In the revised manuscript the authors addressed my comments with additional experiments and their new results and discussion strongly increased the impact of this excellent work.

Stanislav Kopriva

Reviewer #2:

Remarks to the Author:

All requested changes have been performed.

Frédéric Marsolais

Reviewer #4:

Remarks to the Author:

The REVISED manuscript describes candidate genes (and their relative transcript abundance) encoding four enzymes leading to dimethylsulfoniopropionate (DMSP) synthesis in multiple tissues of *Spartina anglica*, an invasive salt marsh grass. The authors also verified the enzyme activities of the corresponding recombinant enzymes. The authors also surveyed the accumulation of DMSP in situ in a natural saltmarsh setting and correlated such accumulation with sulfate availability, not NaCl exposure. The authors went on to survey 42 different plant species for DMSP accumulation to show that such high accumulation is rare within the plant kingdom. Lastly, the authors also demonstrate that DMSP uptake by roots and overexpression of the DMSP genes led to increased DMSP accumulation and improved salinity and drought tolerance in tomato, *Arabidopsis*, and tobacco (transiently).

Overall, the manuscript is greatly improved and the authors have addressed successfully the comments raised by the reviewers. However, claims of precedence as the identification of the first DMSP synthesis genes in plants should be removed in case the authors are incorrect. This is a typical policy for many top plant journals.